# A protein quality control pathway at the mitochondrial outer membrane

**Meredith B Metzger\*, Jessica L Scales, Mitchell F Dunklebarger, Jadranka Loncarek, Allan M Weissman\***

Laboratory of Protein Dynamics and Signaling, Center for Cancer Research, National Cancer Institute at Frederick, Frederick, United States

**Abstract** Maintaining the essential functions of mitochondria requires mechanisms to recognize and remove misfolded proteins. However, quality control (QC) pathways for misfolded mitochondrial proteins remain poorly defined. Here, we establish temperature-sensitive (ts-) peripheral mitochondrial outer membrane (MOM) proteins as novel model QC substrates in *Saccharomyces cerevisiae*. The ts- proteins sen2-1HA[ts] and sam35-2HA[ts] are degraded from the MOM by the ubiquitin-proteasome system. Ubiquitination of sen2-1HA[ts] is mediated by the ubiquitin ligase (E3) Ubr1, while sam35-2HA[ts] is ubiquitinated primarily by San1. Mitochondria-associated degradation (MAD) of both substrates requires the SSA family of Hsp70s and the Hsp40 Sis1, providing the first evidence for chaperone involvement in MAD. In addition to a role for the Cdc48-Npl4-Ufd1 AAA-ATPase complex, Doa1 and a mitochondrial pool of the transmembrane Cdc48 adaptor, Ubx2, are implicated in their degradation. This study reveals a unique QC pathway comprised of a combination of cytosolic and mitochondrial factors that distinguish it from other cellular QC pathways.

## Introduction

Proper protein folding is essential for organelle and cell homeostasis. Proteins may fail to achieve or retain their functional conformations due to genetic or environmental insults and cells have evolved elaborate protein quality control (QC) mechanisms to combat misfolding. QC systems recognize and refold misfolded proteins and, when not possible, sequester or degrade them to prevent their often deleterious accumulation (*Kevei et al., 2017*). Such degradation frequently occurs via the ubiquitin-proteasome system (UPS), where substrates are modified with ubiquitin leading to their destruction by the 26S proteasome (*Metzger et al., 2012*). Ubiquitin modification requires the action of ubiquitin-activating enzyme (E1), followed by either sequential or concerted activity of ubiquitin-conjugating enzymes (E2) and substrate-specific ubiquitin ligases (E3). Multiple rounds of ubiquitination result in polyubiquitin chains, which can serve as targeting signals for recognition and degradation by 26S proteasomes (*Akutsu et al., 2016*; *Thrower et al., 2000*). Other classes of factors play critical roles in substrate recognition and in their extraction from associated proteins or membranes post-ubiquitination, as well as in shuttling to proteasomes (*Buchberger et al., 2015*; *Ye et al., 2017*; *Zientara-Rytter and Subramani, 2019*).

Distinct, yet overlapping UPS machinery found at different subcellular locations mediates localized QC of misfolded proteins (*Sontag et al., 2017*). The degradation of misfolded proteins at the ribosome, endoplasmic reticulum (ER), nucleus, inner nuclear envelope, and cytosol is mediated by dedicated UPS machinery (*Boban and Foisner, 2016*; *Brandman and Hegde, 2016*; *Comyn et al., 2014*; *Gamerdinger, 2016*; *Jones and Gardner, 2016*; *Zattas and Hochstrasser, 2015*). Much of what is known about these subcellular QC pathways was uncovered in yeast using model misfolded substrates (*Bays et al., 2001*; *Gardner et al., 2005*; *Huyer et al., 2004*; *Stolz and Wolf, 2012*; *Vashist and Ng, 2004*) and has served as the basis for characterizing mammalian degradation

**\*For correspondence:**
metzgermb@mail.nih.gov (MBM);
weissmaa@mail.nih.gov (AMW)

**Competing interests:** The authors declare that no competing interests exist.

**eLife digest** Proteins are molecules that need to fold into the right shape to do their job. If proteins lose that shape, not only do they stop working but they risk clumping together and becoming toxic, potentially leading to disease. Fortunately, the cell has quality control systems that normally detect and remove misfolded proteins before they can cause damage to the cell. First, sets of proteins known as chaperones recognize the misfolded proteins, and then another class of proteins attaches a molecular tag, known as ubiquitin, to the misshapen proteins. When several ubiquitin tags are attached to a protein, forming chains of ubiquitin, it is transported to a large molecular machine within the cell called the proteasome. The proteasome unravels the protein and breaks it down into its constituent building blocks, which can then be used to create new proteins.

Proteins are found throughout the different compartments of the cell and quality control processes have been well-studied in some parts of the cell but not others. Metzger et al. have now revealed how the process works on the surface of mitochondria, the compartment that provides the cell with most of its energy. To do this, they used baker's yeast, a model laboratory organism that shares many fundamental properties with animal cells, but which is easier to manipulate genetically. The quality control process was studied using two mitochondrial proteins that had been mutated to make them sensitive to changes in temperature. This meant that, when the temperature increased from 25°C to 37°C, these proteins would begin to unravel and trigger the clean-up operation. This approach has been used previously to understand the quality control processes in other parts of the cell.

By removing different quality control machinery in turn from the yeast cells, Metzger et al. could detect which were necessary for the process on mitochondria. This showed that there were many similarities with how this process happen in other parts of the cell but that the precise combination of chaperones and enzymes involved was distinct. Furthermore, when the proteasome was not working, the misfolded proteins remained on the mitochondria, showing that they are not transported to other parts of the cell to be broken down. In the future, understanding this process could help to find potential drug targets for mitochondrial diseases. The next steps will be to see how well these findings apply to human and other mammalian cells.

pathways. The most extensively studied organelle-based QC pathway is ER-associated degradation (ERAD). Early steps in ERAD pathways, such as substrate recognition and ubiquitination, are distinct for different substrates and defined (at least in yeast) by the location of the misfolded domain (*Carvalho et al., 2006*; *Huyer et al., 2004*; *Preston and Brodsky, 2017*; *Ruggiano et al., 2014*). Generally, pathways converge post-ubiquitination, where the Cdc48-Npl4-Ufd1 AAA-ATPase complex and its associated co-factors facilitate unfolding and/or extraction of substrates from their natural environments prior to targeting to 26S proteasomes for degradation (*Olszewski et al., 2019*; *Wolf and Stolz, 2012*).

Mitochondrial proteins are subject to ongoing oxidative insults that can result in damage, misfolding, and dysfunction. Mitochondria require mechanisms to eliminate these proteins to maintain organellar integrity and essential functions (*Voos et al., 2016*). In mammalian cells, the well-characterized Parkin and PINK1 ubiquitin-dependent mitophagy pathway removes portions of, or entire, damaged mitochondria (*McWilliams and Muqit, 2017*; *Pickles et al., 2018*). In yeast, however, mitophagy appears to function primarily to adapt to metabolic changes, rather than in protein QC (*Fukuda and Kanki, 2018*; *Kanki et al., 2015*). Furthermore, there is little evidence indicating that mitophagy is ubiquitin-dependent, in accordance with the absence of obvious yeast orthologues of Parkin or PINK1 (*Belgareh-Touzé et al., 2017*; *Tan et al., 2016*).

In contrast to mitophagy, mitochondria-associated degradation (MAD) pathways for individual misfolded or damaged mitochondrial proteins are not as well-established in mammals or yeast. Proteases resident to the mitochondrial matrix, inner membrane (IM), and intermembrane space (IMS) can act on damaged or aggregated proteins in these compartments (*Bohovych et al., 2015*). While there is no evidence for proteasomes inside mitochondria, the mitochondrial outer membrane (MOM) is fully accessible to cytosolic proteasomes. In fact, the UPS is known to play a critical role in mitochondrial morphology, dynamics, inheritance, and in the degradation of import-deficient

mitochondrial IMS proteins and mitochondria-mislocalized tail-anchored proteins (*Altmann and Westermann, 2005*; *Bragoszewski et al., 2017*; *Cohen et al., 2008*; *Fisk and Yaffe, 1999*; *Goodrum et al., 2019*; *Matsumoto et al., 2019*; *Rinaldi et al., 2008*). A limited number of MOM proteins have also been identified as specific UPS targets in yeast: Fzo1, Mdm12, Mdm34, Msp1, and Tom70 (*Belgareh-Touzé et al., 2017*; *Cohen et al., 2008*; *Fritz et al., 2003*; *Ota et al., 2008*; *Wu et al., 2016*). All of these are native (*i.e.* non-misfolded) proteins, whose ubiquitination and/or degradation may be critical for homeostasis and not obviously related to QC. Of these, only Fzo1 has been examined in detail. The regulated recognition and ubiquitination of this mitofusin by SCF^Mdm30 (Skp1-Cullin-F-box E3 with F-box protein Mdm30) and its subsequent proteasomal degradation are integral to the process of MOM fusion (*Cohen et al., 2011*; *Cohen et al., 2008*; *Escobar-Henriques et al., 2006*). However, for the few other MOM proteins where ubiquitination has been analyzed, the E3 Rsp5 has been implicated (*Belgareh-Touzé et al., 2017*; *Goodrum et al., 2019*; *Kowalski et al., 2018*; *Wu et al., 2016*). The involvement of Cdc48 co-factors, Vms1 and Doa1, has also been both reported and disputed for particular MOM proteins (*Chowdhury et al., 2018*; *Esaki and Ogura, 2012*; *Heo et al., 2010*; *Wu et al., 2016*). The degradation of tail-anchored proteins mislocalized to the MOM uniquely requires the AAA-ATPase Msp1 for extraction from the MOM prior to transfer to the ER where they are degraded by ERAD machinery (*Matsumoto et al., 2019*; *Okreglak and Walter, 2014*; *Wohlever et al., 2017*). Thus, a universal MAD pathway for MOM proteins has not been described and many steps in substrate degradation remain unexamined. Similarly, UPS components acting on misfolded MOM proteins have yet to be investigated.

In this study, we establish temperature-sensitive (ts-) peripheral MOM proteins (sam35-2HA^ts and sen2-1HA^ts) as QC substrates. The ts- nature of these substrates, coupled with their tight association with the MOM, enabled study of their mitochondrial degradation without concerns of mislocalization to the cytosol or elsewhere. We utilize these substrates to define a MAD pathway for non-native proteins. The proteasomal degradation of these MAD QC substrates occurs at the MOM and requires specific cytosolic and mitochondrial UPS components, most of which are conserved in higher eukaryotes. Our results reveal a requirement for factors not previously implicated in the degradation of native MAD substrates and the combination of components identified defines a distinct QC pathway.

## Results

### Identification of novel thermosensitive substrates for mitochondrial quality control

Our knowledge of protein QC in the ER, cytosol, and nucleus derives in part from the study of proteins that undergo temperature-dependent misfolding and degradation (*Biederer et al., 1996*; *Gardner et al., 2005*; *Khosrow-Khavar et al., 2012*; *Ravid et al., 2006*; *Wang and Prelich, 2009*). To elucidate how misfolded mitochondrial proteins are targeted for destruction, we exploited two previously-identified yeast ts- alleles, *sam35-2* and *sen2-1* (*Li et al., 2011*; *Milenkovic et al., 2004*; *Winey and Culbertson, 1988*), whose degradation has not been assessed. Sam35 and Sen2 are essential MOM proteins. Sam35 is the substrate receptor of the MOM-embedded multiprotein sorting and assembly machinery (SAM) complex required for assembly of β-barrel proteins into the MOM (*Chan and Lithgow, 2008*; *Kozjak et al., 2003*; *Kutik et al., 2008*; *Milenkovic et al., 2004*). Although Sam35 contains no apparent membrane spans, it is tightly embedded at the MOM via the Sam50 protein (*Kutik et al., 2008*). Sen2 provides endonuclease activity for the multi-subunit tRNA splicing endonuclease complex and also cleaves a mitochondria-targeted non-stop mRNA (*Ho et al., 1990*; *Tsuboi et al., 2015*; *Winey and Culbertson, 1988*). The tRNA splicing complex resides on the MOM in yeast, with Sen2 potentially anchoring it to the membrane via a hydrophobic segment (*Trotta et al., 1997*; *Yoshihisa et al., 2003*).

We first determined that sam35-2^ts and sen2-1^ts encoded full-length proteins. Each contained multiple missense mutations (*Figure 1—figure supplement 1A*) that likely account for the phenotypes reported (*Li et al., 2011*; *Milenkovic et al., 2004*; *Winey and Culbertson, 1988*). To facilitate detection, sequence encoding an HA tag was added to the C-terminus of each to generate sam35-2HA^ts and sen2-1HA^ts. Genome-integrated versions of sam35-2HA^ts and sen2-1HA^ts support cell viability at the permissive (25°C) but not non-permissive temperature (37°C; *Figure 1A*), consistent with

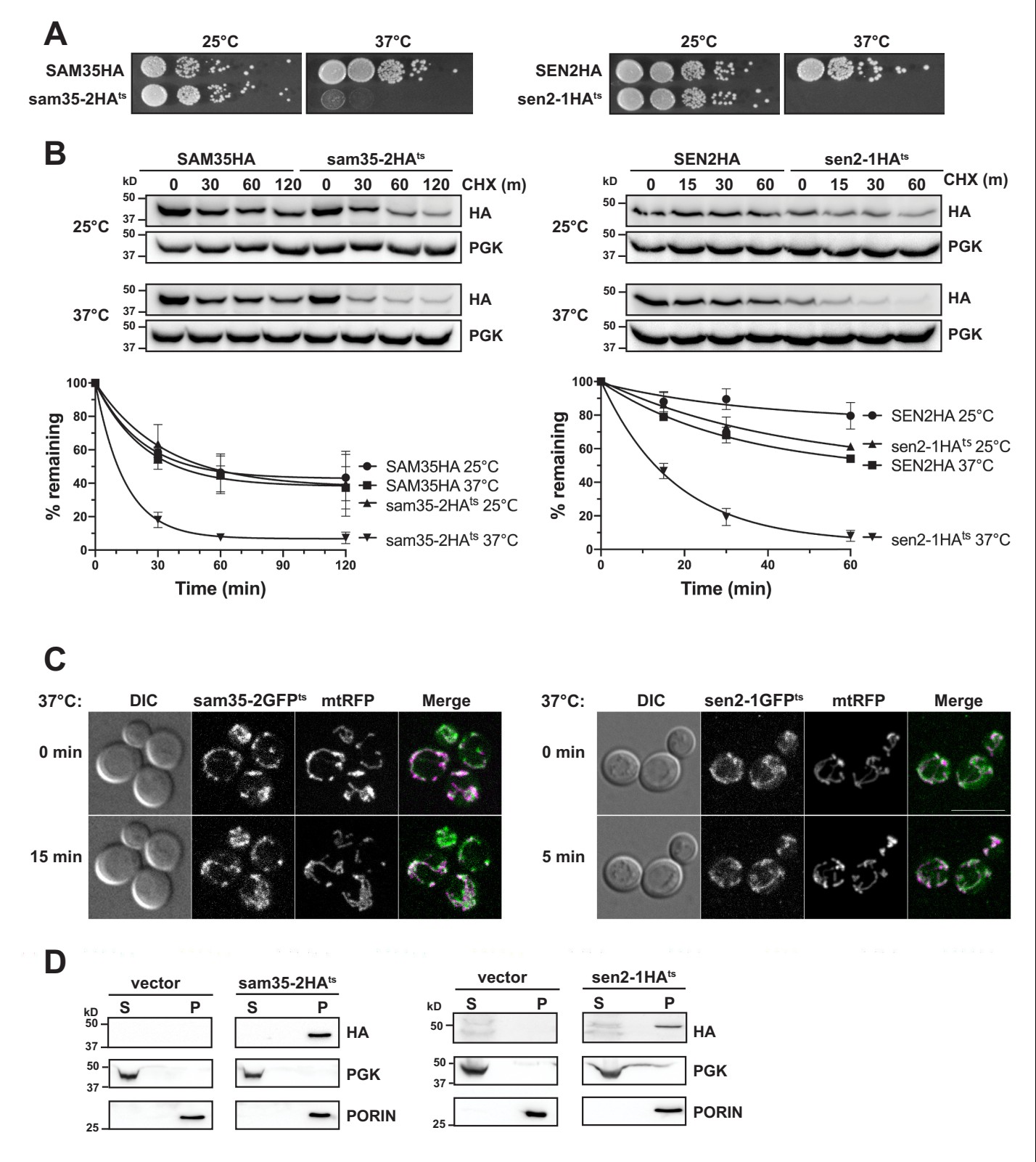

**Figure 1.** The temperature sensitive proteins sam35-2HA[ts] and sen2-1HA[ts] are novel thermosensitive substrates for mitochondrial quality control. (**A**) Spot growth assay of cells expressing chromosomal SAM35HA, sam35-2HA[ts], SEN2HA, or sen2-1HA[ts] (yMM36, 37, 40, and 41, respectively) at permissive (25˚) or non-permissive (37˚) temperatures. (**B**) Wild type (WT; WCG4a) yeast were treated with cycloheximide (CHX) at 25˚C or 37˚C and analyzed at the indicated times to assess the degradation of centromeric (CEN) plasmid-expressed SAM35HA, sam35-2HA[ts], SEN2HA, or sen2-1HA[ts]

*Figure 1 continued on next page*

*Figure 1 continued*

(pMM158, 157, 159, 160, respectively). The ts- proteins were detected by immunoblotting with HA antibody. Phosphoglycerate kinase (PGK) served as a protein loading control. Graphed below is the mean and standard deviation (SD) of the PGK-normalized HA signal at each time point for three biological replicates. (C) Live-cell microscopy analysis of agarose-embedded WT cells (WCG4a) co-expressing a mitochondrial-matrix targeted RFP (mtRFP; pMD12) and either sam35-2GFP$^{ts}$ (pMD1) or sen2-1GFP$^{ts}$ (pMD4) at the indicated times after temperature shift to 37°C. CHX was also added at 0 min, although CHX diffusion through agarose is likely problematic. 'Merge' of GFP (green) and RFP (magenta) channels and differential interference contrast (DIC) are shown; Scale bar = 10 µm. (D) Lysates of spheroplasted yeast from the strains used in B were fractionated at 12,000$x$g at 37°C into mitochondrial pellet (P) and post-mitochondrial supernatant (S). Fractions were subject to immunoblotting with antibodies to HA, PGK (cytosolic protein control), and PORIN (mitochondrial protein control).

The online version of this article includes the following source data and figure supplement(s) for figure 1:

**Source data 1.** Quantifications of cycloheximide chases.
**Figure supplement 1.** The temperature sensitive proteins sam35-2HA$^{ts}$ and sen2-1HA$^{ts}$ are novel thermosensitive substrates for mitochondrial quality control.

the previously-described untagged versions (*Li et al., 2011*; *Milenkovic et al., 2004*; *Winey and Culbertson, 1988*).

To determine whether their ts- phenotypes are indicative of protein misfolding that may lead to instability, we assessed sam35-2HA$^{ts}$ and sen2-1HA$^{ts}$ turnover by cycloheximide (CHX) chase when expressed from centromeric (CEN) yeast plasmids. Cells were grown at 25°C until the addition of CHX at time 'zero', when the temperature was either maintained at 25°C or raised to 37°C. The addition of CHX prior to raising the temperature minimizes the contribution of newly synthesized ts- proteins that may misfold prior to reaching the mitochondria. We find that increasing the temperature to 37°C results in dramatic destabilization of the ts- proteins relative to the WT proteins (*Figure 1B*). Chromosomal versions behaved similarly (*Figure 1—figure supplement 1B*). Thus, the ts- nature of these alleles is likely attributable to misfolding leading to destabilization at the non-permissive temperature.

Next, agarose-embedded cells expressing GFP-tagged sam35-2$^{ts}$ and sen2-1$^{ts}$ were used to assess whether these non-transmembrane MOM proteins remain mitochondrially-associated following the increase to the non-permissive temperature. For this, the temperature was raised to 37°C for 5 min (sen2-1GFP$^{ts}$) or 15 min (sam35-2GFP$^{ts}$), times at which sam35-2GFP$^{ts}$ and sen2-1GFP$^{ts}$ were already destabilized by CHX chase (*Figure 1—figure supplement 1C*). Co-expression of sam35-2GFP$^{ts}$ and sen2-1GFP$^{ts}$ with a mitochondrial marker (mtRFP) demonstrated that the ts- proteins remain mitochondrial after the temperature increase (*Figure 1C*).

To bypass difficulties associated with imaging these low abundance ts- proteins, we isolated mitochondria by subcellular fractionation and assessed the localization of destabilized sam35-2HA$^{ts}$ and sen2-1HA$^{ts}$. Using the same conditions as for microscopy, we found that the HA-tagged ts- proteins are destabilized upon the addition of CHX (*Figure 1—figure supplement 1D*) and that protein turnover was still evident in the spheroplasted yeast used for fractionation (WT; *Figure 1—figure supplement 1E*). Consistent with our microscopy, sam35-2HA$^{ts}$ and sen2-1HA$^{ts}$ fractionated almost exclusively to the mitochondrial pellet at 37°C (P; *Figure 1D*). A similar fractionation pattern was observed with chromosomal HA-tagged ts- alleles (*Figure 1—figure supplement 1F*). Therefore, sam35-2HA$^{ts}$ and sen2-1HA$^{ts}$ are destabilized at the non-permissive temperature yet remain mitochondrially-localized, validating their use as model MAD QC substrates.

## The degradation of MAD QC substrates requires the ubiquitin-proteasome system

Since proteasomes have broad roles in mitochondrial homeostasis and morphology, we assessed whether the degradation of these MOM ts- proteins was dependent on proteasome function. The turnover of sam35-2HA$^{ts}$ and sen2-1HA$^{ts}$ was assessed at the non-permissive temperature in strains containing conditional mutations in the 20S core (*pre1-1 pre2-2*; *Figure 2A*) or the 19S cap (*cim3-1*; *Figure 2B*) of the 26S proteasome. Both ts- proteins were substantially stabilized in these mutants relative to isogenic WT strains. In contrast, vacuolar or mitochondrial resident proteases were not required for sam35-2HA$^{ts}$ or sen2-1HA$^{ts}$ turnover (*Figure 2—figure supplement 1A and B*).

After establishing that the stabilization of the ts- proteins in *pre1-1 pre2-2* spheroplasts mirrors that observed in intact cells (*Figure 1—figure supplement 1E*), we assessed whether sam35-2HA$^{ts}$

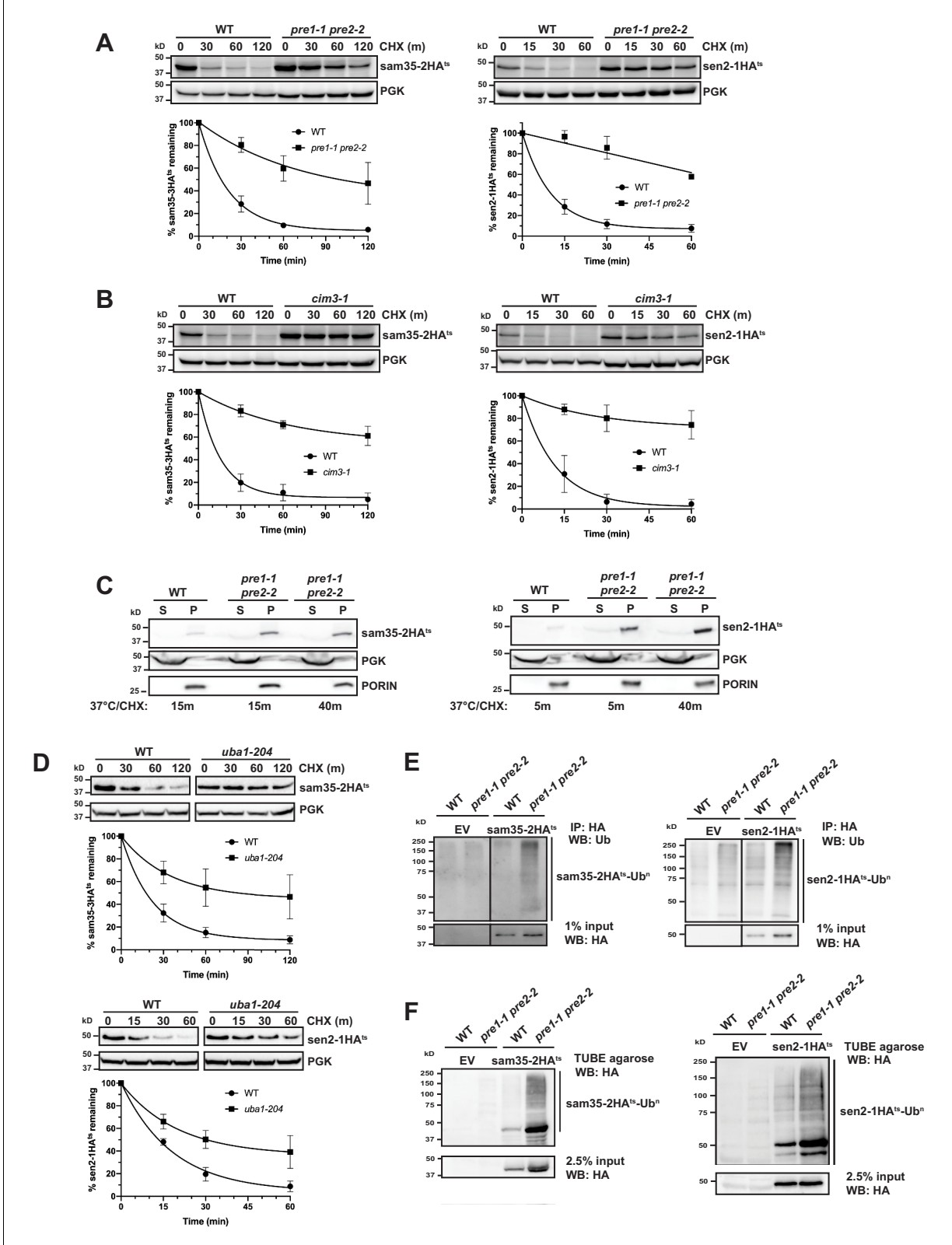

**Figure 2.** The degradation of MAD QC substrates requires the ubiquitin-proteasome system. (A, B) CHX chase for the indicated times at 37°C assessing the turnover of sam35-2HA^ts or sen2-1HA^ts (pMM157 or 160, respectively) in WT (WCG4a) and *pre1-1 pre2-2* proteasome mutant (WCG4-11/21a) cells (A) or WT (CIM) and *cim3-1* proteasome mutant cells (B). Proteins were detected by immunoblotting. Graphed below is the mean and SD of the PGK-normalized HA signal at each time point for three biological replicates. (C) Lysates from the strains used in A were fractionated at 12,000x*g* into

*Figure 2 continued on next page*

*Figure 2 continued*

mitochondrial pellets (P) and post-mitochondrial supernatants (S) after incubation at 37°C for the indicated times. Fractions were subject to immunoblotting with antibodies to HA, PGK, and PORIN. (D) CHX chase for the indicated times at 37°C assessing the turnover of sam35-2HA$^{ts}$ or sen2-1HA$^{ts}$ (pMM157 or 160, respectively) in a *uba1-204* strain relative to its isogenic WT strain. (E) Ubiquitination of sam35-2HA$^{ts}$ and sen2-1HA$^{ts}$ was assessed by immunoprecipitation (IP) from lysates of the strains used in A with anti-HA agarose, followed by immunoblotting with ubiquitin antibodies. 1% of IP input lysate was reserved and also analyzed by immunoblotting. (F) Ubiquitination of sam35-2HA$^{ts}$ and sen2-1HA$^{ts}$ was assessed by IP from lysates of the strains used in A using tandem ubiquitin-binding entities (TUBE) agarose, followed by immunoblotting with HA antibody. 2.5% of the TUBE input lysate was reserved and analyzed by immunoblotting.

The online version of this article includes the following source data and figure supplement(s) for figure 2:

**Source data 1.** Quantifications of cycloheximide chases.

**Figure supplement 1.** The degradation of MAD QC substrates requires the ubiquitin-proteasome system.

and sen2-1HA$^{ts}$ remain mitochondrial when proteasome function is impaired. Even after 40 min at the non-permissive temperature, the majority of both sam35-2HA$^{ts}$ and sen2-1HA$^{ts}$ accumulated in the 12,000*xg* mitochondrial pellet (*Figure 2C*). Extraction of both proteins from this pellet fraction by sodium carbonate excluded the possibility that this represents aggregated protein (*Figure 2—figure supplement 1C*). This crude mitochondrial pellet also contains ~20% of total ER as assessed by measuring levels of the integral ER membrane protein Cue1 (*Figure 2—figure supplement 1D*). For this reason, we wished to exclude the possibility that these substrates were being translocated to the ER for degradation. Such a pathway has recently been described for tail-anchored proteins that are mislocalized to mitochondria (*Matsumoto et al., 2019*). Further purification of the crude 12,000*xg* mitochondrial pellet isolated from *pre1-1 pre2-2* cells via sucrose gradient ultracentrifugation removes ~90% of contaminating ER, leaving only ~2% of total ER in these purified mitochondria (Cue1; *Figure 2—figure supplement 1E*). On the other hand, this purified mitochondrial fraction retains ~70% of the ts- proteins. As peripheral membrane proteins could dissociate from mitochondria during manipulation, 70% may be an underrepresentation of the mitochondria-associated pool in vivo. Thus, these data indicate that sam35-2HA$^{ts}$ and sen2-1HA$^{ts}$ primarily accumulate at the mitochondria when their proteasomal degradation is blocked.

Consistent with being UPS substrates, stabilization of both sam35-2HA$^{ts}$ and sen2-1HA$^{ts}$ was observed in a ts- mutant of the ubiquitin-activating enzyme (*uba1-204*; *Figure 2D*; *Ghaboosi and Deshaies, 2007*), indicating that ubiquitination is required for their degradation. Moreover, ubiquitin-modified forms of sam35-2HA$^{ts}$ and sen2-1HA$^{ts}$ were evident when proteasome function was inhibited (*Figure 2E and F* and see *Figure 2—figure supplement 1F* for an additional control), as indicated by characteristic higher molecular weight bands and a smear representing highly ubiquitinated species (*Emmerich and Cohen, 2015*). By comparison, ubiquitinated forms of WT SAM35HA and SEN2HA are much less prevalent (*Figure 2—figure supplement 1G*). We further assessed whether ubiquitin-modified forms of the ts- proteins were localized to mitochondria under conditions of proteasome impairment. Ubiquitinated sam35-2HA$^{ts}$ and sen2-1HA$^{ts}$ both accumulated largely in the mitochondrial pellet fraction (P12,000x*g*; *Figure 2—figure supplement 1H*) and to a lesser extent in the soluble fraction, correlating with the relative distributions of unmodified species. Taken together, these data establish roles for ubiquitination and proteasomal degradation in the turnover of sam35-2HA$^{ts}$ and sen2-1HA$^{ts}$ at the MOM.

## Distinct E3 ubiquitin ligases act on sam35-2HA$^{ts}$ and sen2-1HA$^{ts}$

To date, the only ubiquitin ligases known to act on mitochondrial proteins in yeast are SCF$^{Mdm30}$ and Rsp5 (*Belgareh-Touzé et al., 2017*; *Cohen et al., 2008*; *Escobar-Henriques et al., 2006*; *Kowalski et al., 2018*; *Wu et al., 2016*). However, the known mitochondrial substrates for these E3s are native proteins. We assessed whether these E3s also play a role in MAD of QC substrates and found that strains lacking either Mdm30 or Rsp5 function did not affect sam35-2HA$^{ts}$ or sen2-1HA$^{ts}$ stability (*Figure 3—figure supplement 1A and B*). To identify factors involved in targeting these ts-MAD substrates, we screened a yeast deletion library consisting of non-essential known or putative UPS components (*Ravid and Hochstrasser, 2007*) by CHX chase. Interestingly, we found that *san1Δ* impaired turnover of sam35-2HA$^{ts}$ (*Figure 3A*, middle panel), while *ubr1Δ* impaired turnover of sen2-1HA$^{ts}$ (*Figure 3B*, middle panel). San1 has been implicated in QC of cytosolic and nuclear proteins (*Gardner et al., 2005*; *Heck et al., 2010*). Ubr1 is the E3 that ubiquitinates N-end rule

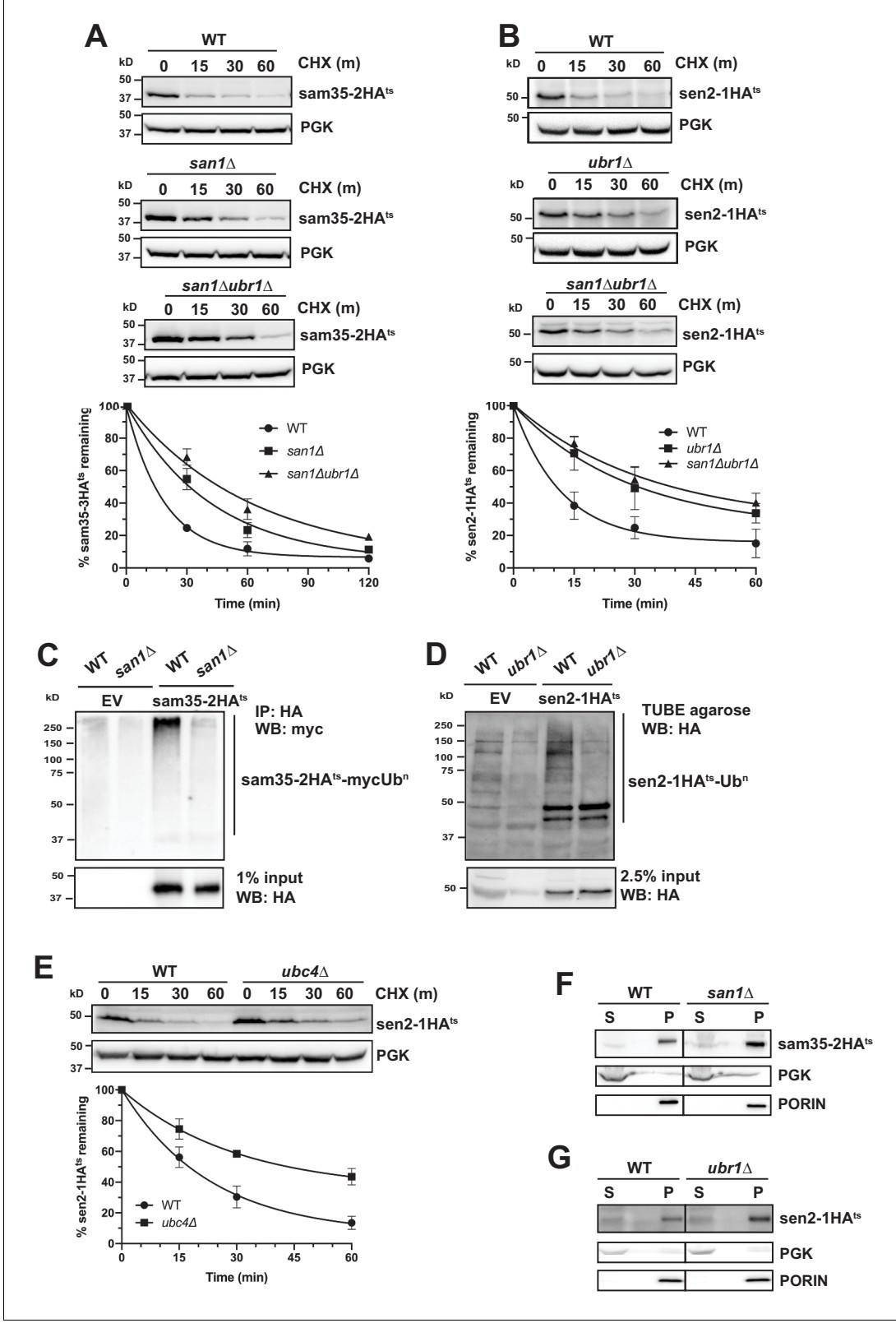

**Figure 3.** Distinct E3 ubiquitin ligases act on sam35-2HA[ts] and sen2-1HA[ts]. (**A**) CHX chase for the indicated times at 37°C assessing the turnover of sam35-2HA[ts] (pMM157) in WT (BY4741), *san1Δ*, and *san1Δ ubr1Δ* (SM5770) cells. Proteins were detected by immunoblotting. Graphed below is the mean and SD of the PGK-normalized HA signal at each time point for three biological replicates. (**B**) CHX chase for the indicated times at 37°C assessing the turnover of sen2-1HA[ts] (pMM160) in WT (BY4741), *ubr1Δ* (yMM149), and *san1Δ ubr1Δ* (SM5770) cells, as in A. (**C**) Ubiquitination of sam35-

*Figure 3 continued on next page*

Figure 3 continued

2HA[ts] was assessed by IP with anti-HA agarose of lysates from WT (BY4741) and *san1Δ* cells expressing myc-Ub (pSM3666) and either empty vector (EV; pRS315) or sam35-2HA[ts] (pMM157), followed by immunoblotting with c-myc antibody. 1% of IP input lysate was reserved and analyzed for sam35-2HA[ts] by immunoblotting. (D) Ubiquitination of sen2-1HA[ts] was assessed by IP of lysates from WT and *ubr1Δ* strains expressing either EV (pRS315) or sen2-1HA[ts] (pMM160) using TUBE agarose, followed by immunoblotting with HA antibody. 2.5% of the TUBE input lysate was reserved and analyzed by immunoblotting for sen2-1HA[ts]. (E) CHX chase for the indicated times at 37°C assessing the turnover of sen2-1HA[ts] (pMM160) in *ubc4Δ* cells compared to isogenic WT (BY4741). (F) Lysates from the WT and *san1Δ* strains used in A expressing sam35-2HA[ts] (pMM157) were fractionated at 12,000×g at 37°C into mitochondrial pellets (P) and post-mitochondrial supernatants (S). Fractions were subject to immunoblotting with antibodies to HA, PGK, and PORIN. (G) Lysates from the WT and *ubr1Δ* strains used in B expressing sen2-1HA[ts] (pMM160) were fractionated at 12,000×g at 37°C into mitochondrial pellets (P) and post-mitochondrial supernatants (S). Fractions were subject to immunoblotting with antibodies to HA, PGK, and PORIN.

The online version of this article includes the following source data and figure supplement(s) for figure 3:

Source data 1. Quantifications of cycloheximide chases.
Figure supplement 1. Distinct E3 ubiquitin ligases act on sam35-2HA[ts] and sen2-1HA[ts].

substrates, but also plays a prominent role in the turnover of cytosolic proteins and, in some instances, may also degrade ER proteins (*Bartel et al., 1990*; *Comyn et al., 2016*; *Eisele and Wolf, 2008*; *Heck et al., 2010*; *Khosrow-Khavar et al., 2012*; *Rao et al., 2001*; *Scazzari et al., 2015*; *Stolz et al., 2013*; *Summers et al., 2013*). Since there are examples where these two E3s functionally interact, we assessed turnover in a *san1Δ ubr1Δ* double deletion strain. While *ubr1Δ* alone did not affect sam35-2HA[ts] turnover (*Figure 3—figure supplement 1C*), loss of Ubr1 in conjunction with *san1Δ* led to greater stabilization than loss of San1 alone (*Figure 3A*, lower panel), indicating that Ubr1 can target sam35-2HA[ts] in the absence of San1. Loss of San1, alone or in conjunction with *ubr1Δ*, did not affect the stability of sen2-1HA[ts] (*Figure 3—figure supplement 1D* and *Figure 3B*, lower panel). Notably, the polyubiquitination of sam35-2HA[ts] (*Figure 3C*) and sen2-1HA[ts] (*Figure 3D*) was greatly reduced when San1 or Ubr1, respectively, were deleted. Neither ts- protein was entirely stabilized by *san1Δ ubr1Δ*, suggesting that other E3s, perhaps with overlapping specificity or essential functions, may also act on these substrates. Likewise, loss of any single E2 did not stabilize sam35-2HA[ts] (data not shown). Loss of Ubc4, an E2 known to function with Ubr1 for select substrates, stabilized sen2-1HA[ts] to a similar degree as *ubr1Δ* (*Figure 3E*).

To confirm that Ubr1 and San1 can act on ts- proteins at the MOM, we examined the localization of sam35-2HA[ts] and sen2-1HA[ts] by fractionation in *san1Δ* and *ubr1Δ*, respectively. In these strains, both ts- proteins still accumulated almost exclusively in the pellet fraction (*Figure 3F and G*). These results establish that San1 and Ubr1 can function as components of the MAD machinery for proteasome-mediated degradation of sam35-2HA[ts] and sen2-1HA[ts], respectively.

## Cytosolic chaperones are required for MAD QC substrate degradation

The cytosolic SSA family of Hsp70 chaperones (Ssa1, Ssa2, Ssa3, and Ssa4) is broadly involved in protein folding and refolding, but also plays a role in the degradation of several San1 and Ubr1 QC substrates (*Guerriero et al., 2013*; *Heck et al., 2010*; *Prasad et al., 2010*; *Stolz et al., 2013*). During protein QC, these chaperones appear to prevent protein aggregation and facilitate ubiquitination (*Huyer et al., 2004*; *Metzger et al., 2008*; *O'Donnell et al., 2017*; *Park et al., 2007*; *Scazzari et al., 2015*; *Stolz et al., 2013*). However, they have not been implicated in MAD. Interestingly, both sam35-2HA[ts] and sen2-1HA[ts] were strongly stabilized in a conditional yeast strain where three SSA family members are deleted and the fourth (*SSA1*) is ts- (*ssa1-45[ts]*; *ssa1-45[ts] ssa2Δ ssa3Δ ssa4Δ*) relative to the control strain (*SSA1*; *SSA1 ssa2Δ ssa3Δ ssa4Δ*; *Figure 4A*).

SSA Hsp70 chaperones can play a role in the sorting of mitochondrial proteins to and into mitochondria (*Deshaies et al., 1988*; *Eliyahu et al., 2012*; *Young et al., 2003*). To rule out the possibility that the stabilization of mitochondrial ts- alleles in the *ssa1-45[ts]* strain is reflective of a defect in mitochondrial targeting, mitochondrial fractionation was performed. Both sam35-2HA[ts] and sen2-1HA[ts] were localized to the pellet fraction (*Figure 4B*), where they were extractable by sodium carbonate (*Figure 4—figure supplement 1A*). Notably, the SSA1 chaperone was also required for ubiquitination: despite the accumulation of unmodified sam35-2HA[ts] and sen2-1HA[ts], ubiquitin-modified forms are dramatically reduced when SSA1 is inactivated (*Figure 4C*). This is consistent with the established QC function for SSA chaperones in engaging substrates prior to their ubiquitination (*Shiber and Ravid, 2014*). In contrast to the ts- proteins, degradation of the native MOM UPS

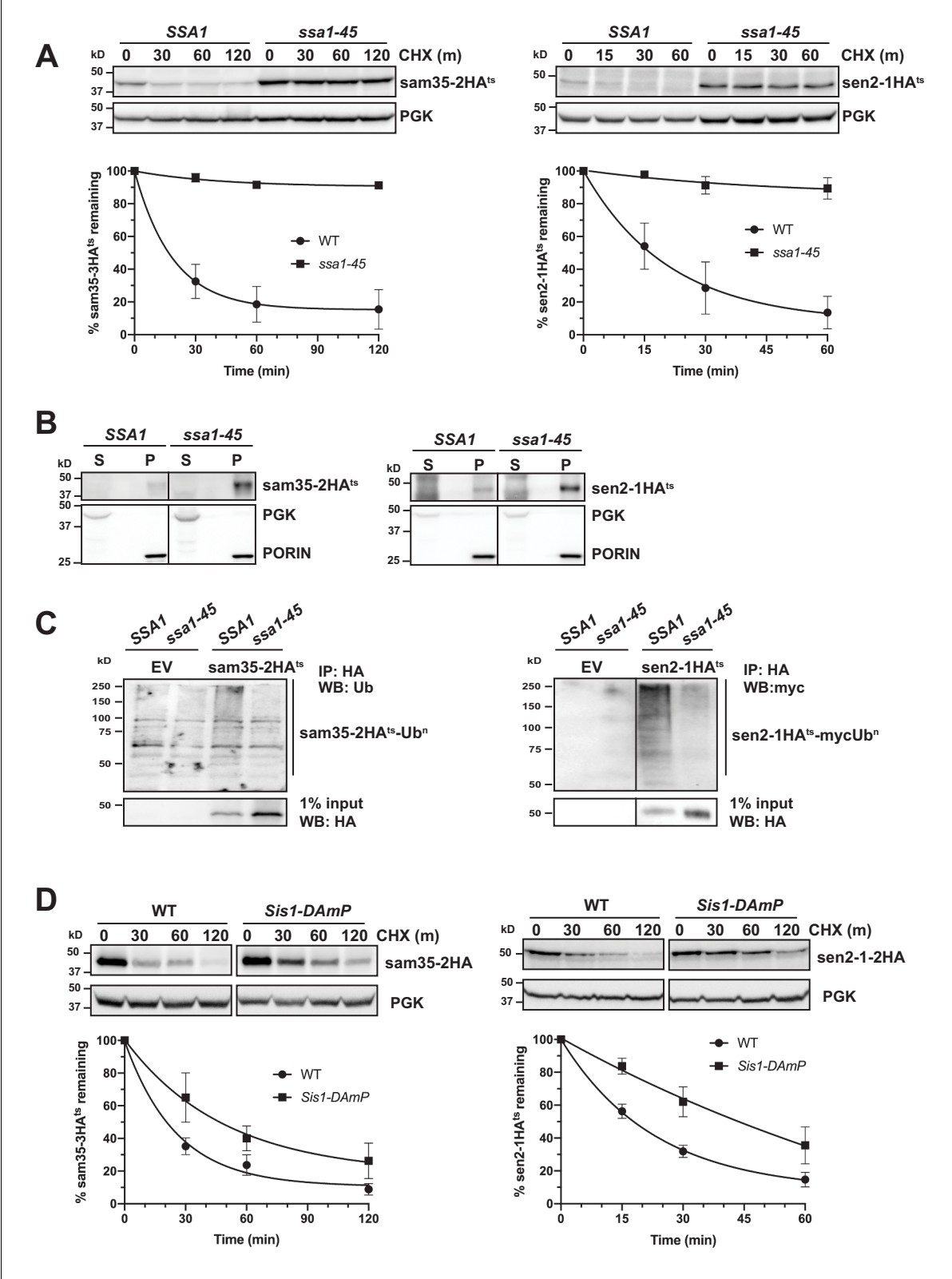

**Figure 4.** Cytosolic chaperones are required for the degradation of sen2-1HA^ts and sam35-2HA^ts. (**A**) CHX chase for the indicated times at 37°C assessing the turnover of sam35-2HA^ts (pMM231) and sen2-1HA^ts (pMM234) in WT (*SSA1*) and *ssa1-45*^ts cells. Proteins were detected by immunoblotting. Graphed below is the mean and SD of the PGK-normalized HA signal at each time point for three biological replicates. (**B**) Lysates from WT and *ssa1-45*^ts strains expressing sam35-2HA^ts (pMM231) or sen2-1HA^ts (pMM234) were fractionated at 12,000x*g* at 37°C into mitochondrial

*Figure 4 continued on next page*

*Figure 4 continued*

pellets (P) and post-mitochondrial supernatants (S). Fractions were subject to immunoblotting with antibodies to HA, PGK, and PORIN. (**C**) Ubiquitination of sam35-2HA$^{ts}$ and sen2-1HA$^{ts}$ was assessed by IP with anti-HA agarose from lysates from WT and *ssa1-45$^{ts}$* cells expressing either empty vector (EV; pRS316), sam35-2HA$^{ts}$ (pMM231), or sen2-1HA$^{ts}$ (pMM234), followed by immunoblotting with either ubiquitin or c-myc antibody. 1% of IP input lysate was reserved and analyzed by immunoblotting. (**D**) CHX chase for the indicated times at 37°C assessing the turnover of sam35-2HA$^{ts}$ (pMM157) and sen2-1HA$^{ts}$ (pMM160) in WT (*yTHC*) and *Sis1-DAmP* cells treated with 10 μg/mL doxycycline for 18 hr at 25°C to decrease Sis1 mRNA abundance prior to the addition of CHX.

The online version of this article includes the following source data and figure supplement(s) for figure 4:

**Source data 1.** Quantifications of cycloheximide chases.

**Figure supplement 1.** Cytosolic chaperones are required for the degradation of sen2-1HA$^{ts}$ and sam35-2HA$^{ts}$.

substrate, Fzo1HA, was largely unaffected by loss SSA function (*Figure 4—figure supplement 1B*). This is consistent with a dichotomy in chaperone requirements between native and non-native MAD substrates.

Hsp40 co-chaperones (J-proteins) stimulate the ATPase activity of Hsp70, which is required for substrate interactions. In particular, Sis1 and Ydj1 play roles in SSA-dependent protein QC throughout the cell (*Heck et al., 2010*; *Lu and Cyr, 1998*; *Prasad et al., 2018*; *Shiber and Ravid, 2014*; *Summers et al., 2013*). Depletion of ~90% of Sis1 protein using Sis1 DAmP cells (*Figure 4—figure supplement 1C*) slowed the turnover of both sam35-2HA$^{ts}$ and sen2-1HA$^{ts}$ (*Figure 4D*), while loss of Ydj1 or its ortholog, Hlj1, did not (*Figure 4—figure supplement 1D*). Recent studies indicate that the degradation of many San1 and Ubr1 cytosolic substrates occurs following nuclear import that is dependent both on Ydj1 and the Hsp70 nucleotide exchange factor (Hsp110) Sse1 (*Prasad et al., 2018*; *Samant et al., 2018*). We did not detect a role for Sse1 in the degradation of sam35-2HA$^{ts}$ and sen2-1HA$^{ts}$ (*Figure 4—figure supplement 1E*). Loss of other factors implicated in San1 and Ubr1 nuclear import-dependent QC, including Hsc82/Hsp82, Sti1, Hsp104, and Dsk2 were also without effect on the degradation of sam35-2HA$^{ts}$ and sen2-1HA$^{ts}$ (*Figure 4—figure supplement 1F*). These results establish that the degradation of the mitochondrial QC substrates is dependent on both Hsp70 and Hsp40 chaperones, but independent of nuclear import.

## The Cdc48-Npl4-Ufd1 complex is required for degradation of MAD QC substrates

The AAA-ATPase Cdc48 plays a broad role in many QC pathways, generally functioning as a protein 'unfoldase' or 'segregase,' while also maintaining protein solubility prior to proteasomal degradation (*Neal et al., 2017*; *Ye et al., 2017*). Cdc48 and its co-factors Npl4 and Ufd1 have been implicated in the UPS-mediated turnover of several yeast MOM proteins (*Cohen et al., 2008*; *Heo et al., 2010*; *Wu et al., 2016*). Both sam35-2HA$^{ts}$ and sen2-1HA$^{ts}$ were stabilized in conditional Cdc48, Npl4, and Ufd1 strains (*Figure 5A*). In contrast, loss of Msp1, another AAA-ATPase that localizes to mitochondria and dislocates mislocalized tail-anchored proteins from the MOM (*Chen et al., 2014*; *Okreglak and Walter, 2014*; *Wohlever et al., 2017*), stabilized neither sam35-2HA$^{ts}$ nor sen2-1HA$^{ts}$ (*Figure 5—figure supplement 1A*).

Given the role of the Cdc48-Npl4-Ufd1 complex as a segregase, we examined whether the ts-proteins accumulated at mitochondria when activity of the complex is compromised, as might be predicted. For both *cdc48-3* and its isogenic WT parental strain, interpretation of fractionation results was clouded by some baseline protein detected in the post-mitochondrial supernatant (*Figure 5B*, left panels). For *npl4-1* and *ufd1-1*, however, both substrates remain primarily mitochondrial (*Figure 5B*, middle and right panels). This indicates that in the absence of a functional Cdc48-Npl4-Ufd1 complex sam35-2HA$^{ts}$ and sen2-1HA$^{ts}$ largely retain their mitochondrial localization.

The Cdc48-Npl4-Ufd1 complex generally binds to ubiquitinated proteins (*Bodnar et al., 2018*; *Park et al., 2005*; *Tsuchiya et al., 2017*), acting downstream of E3-mediated ubiquitination. Accordingly, we detected a relative increase in ubiquitinated sam35-2HA$^{ts}$ and sen2-1HA$^{ts}$ in Cdc48-Npl4-Ufd1 complex mutants (*Figure 5C* and *Figure 5—figure supplement 1B and C*). In ERAD, the UBL- and UBA-containing proteins Rad23 and Dsk2 have been implicated as factors that shuttle ubiquitinated substrates from the Cdc48 complex to proteasomes. Deletion of these or another proteasome shuttling factor, Ddi1, did not affect the degradation of sam35-2HA$^{ts}$ or sen2-1HA$^{ts}$ (*Figure 5—figure supplement 1D*), further distinguishing MAD from ERAD. All together, these results indicate

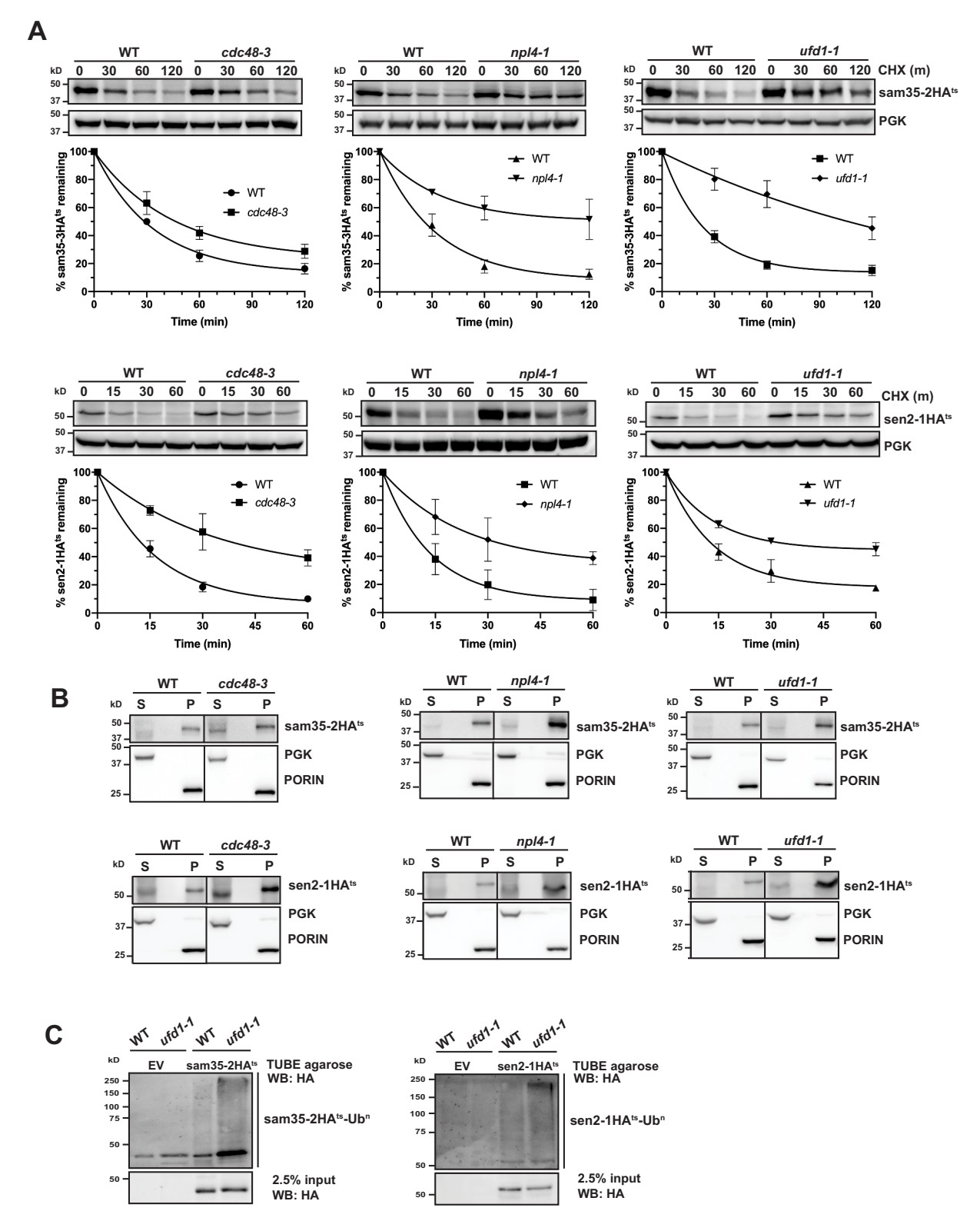

**Figure 5.** The Cdc48-Npl4-Ufd1 complex is required for degradation of MAD substrates. (**A**) CHX chase for the indicated times at 37°C assessing the turnover of sam35-2HA$^{ts}$ (pMM157) or sen2-1HA$^{ts}$ (pMM160) in *cdc48-3*, *npl4-1*, and *ufd1-1* mutant strains compared to isogenic WT strains. Proteins were detected by immunoblotting. Graphed below is the mean and SD of the PGK-normalized HA signal at each time point for three biological replicates. (**B**) Lysates from the strains used in A expressing sam35-2HA$^{ts}$ (pMM157) or sen2-1HA$^{ts}$ (pMM160) were fractionated at 12,000x*g* at 37°C into

*Figure 5 continued on next page*

*Figure 5 continued*

mitochondrial pellets (P) and post-mitochondrial supernatants (S). Fractions were subject to immunoblotting with antibodies to HA, PGK, and PORIN. (C) Ubiquitination of sam35-2HA^ts and sen2-1HA^ts was assessed by IP from lysates of the *ufd1-1* mutant and isogenic WT strain used in A using TUBE agarose, followed by immunoblotting with HA antibody. 2.5% of the TUBE input lysate was reserved and analyzed by immunoblotting.

The online version of this article includes the following source data and figure supplement(s) for figure 5:

**Source data 1.** Quantifications of cycloheximide chases.
**Figure supplement 1.** The Cdc48-Npl4-Ufd1 complex is required for degradation of MAD substrates.

that ubiquitinated mitochondrial QC substrates require the Cdc48-Npl4-Ufd1 complex for efficient proteasomal degradation.

## The Cdc48 complex co-factors Ubx2 and Doa1 are implicated in MAD

The Cdc48-Npl4-Ufd1 complex can be recruited to substrates via Cdc48's interaction with co-factors (*Buchberger et al., 2001*; *Buchberger et al., 2015*; *Wu et al., 2016*). Vms1 and Doa1 are two cytosolic Cdc48 co-factors implicated in mitochondrial homeostasis (*Heo et al., 2010*; *Izawa et al., 2017*; *Nielson et al., 2017*; *Wu et al., 2016*), although their reported involvement in mitochondrial protein turnover, particularly with respect to Fzo1, has been inconsistent (*Chowdhury et al., 2018*; *Esaki and Ogura, 2012*; *Wu et al., 2016*). Loss of Vms1 had no effect on the degradation of sam35-2HA^ts and sen2-1HA^ts (*Figure 6—figure supplement 1A*), while loss of Doa1 had a small, but significant effect (*Figure 6A*).

We next examined deletions of each of the UBX (Ubiquitin-regulatory X) proteins, a family of Cdc48 binding co-factors containing a C-terminal Ub fold (UBX) domain (*Schuberth and Buchberger, 2008*; *Schuberth et al., 2004*). Notably, only the loss of Ubx2 significantly stabilized sam35-2HA^ts and sen2-1HA^ts (*Figure 6B* and *Figure 6—figure supplement 1B*). Ubx2 (as well as its mammalian ortholog, UbxD8) is well-characterized as an ER membrane protein with roles in ERAD and lipid droplet homeostasis (*Kolawa et al., 2013*; *Neuber et al., 2005*; *Schuberth and Buchberger, 2005*; *Wang and Lee, 2012*). We confirmed Ubx2's role in degradation of the mitochondrial ts- proteins by complementation with FLAG-tagged Ubx2 in the *ubx2Δ* strain (*Figure 6C*). Furthermore, redundant functions for Ubx2 and Doa1 were ruled out by a failure of Doa1 overexpression to restore sam35-2HA^ts or sen2-1HA^ts degradation in *ubx2Δ* cells (*Figure 6D*).

As might be predicted given the role of Ubx2 in linking ubiquitinated proteins to the Cdc48 complex, ubiquitinated forms of the ts- substrates accumulated in the absence of Ubx2 (*Figure 6E*). We also detected a physical association between Ubx2-FLAG and the ts- proteins as assessed by co-immunoprecipitation (*Figure 6F*). We see no evidence that loss of Ubx2 affects the already slow turnover of either SAM35HA or SEN2HA (*Figure 6—figure supplement 1C*), and an association between these WT proteins and Ubx2 was also not as pronounced relative to their abundance (*Figure 6—figure supplement 1D*). However, in agreement with recent reports (*Chowdhury et al., 2018*; *Nahar et al., 2020*; *Wu et al., 2016*), we find that loss of Ubx2 and, to a lesser extent, Doa1, stabilizes Fzo1HA, which is a native MOM UPS substrate (*Figure 6—figure supplement 1E and F*). Importantly, we also establish that Fzo1HA physically interacts with Ubx2, accompanied by an increase in ubiquitinated forms (*Figure 6—figure supplement 1G and H*). Our findings are consistent with Ubx2 interacting with UPS-targeted native and misfolded substrates downstream of their ubiquitination to facilitate degradation.

Although Ubx2 is an ER transmembrane protein, one study suggested it may also localize to mitochondria (*Wang and Lee, 2012*), which was recently corroborated (*Mårtensson et al., 2019*). We also confirmed this by co-localization of Ubx2-GFP with both ER (Sec63-RFP) and mitochondrial (mtRFP) markers by microscopy (*Figure 6—figure supplement 1I*). The mitochondrial localization of a portion of Ubx2 was further verified biochemically by isolation of mitochondria largely devoid of co-purifying cytosolic (PGK), ER luminal (CPY), or ER membrane (Cue1) proteins (*Figure 6G*).

As the ER and mitochondria are in close apposition at ER-mitochondrial contact sites, it is possible that ER-localized Ubx2 facilitates mitochondrial protein degradation. Recent studies have also characterized a pathway for degradation of tail-anchored proteins mislocalized to mitochondria that entails Msp1-mediated extraction and subsequent degradation at the ER by ERAD machinery (*Dederer et al., 2019*; *Matsumoto et al., 2019*). To exclude a potential role for the ER in sam35-

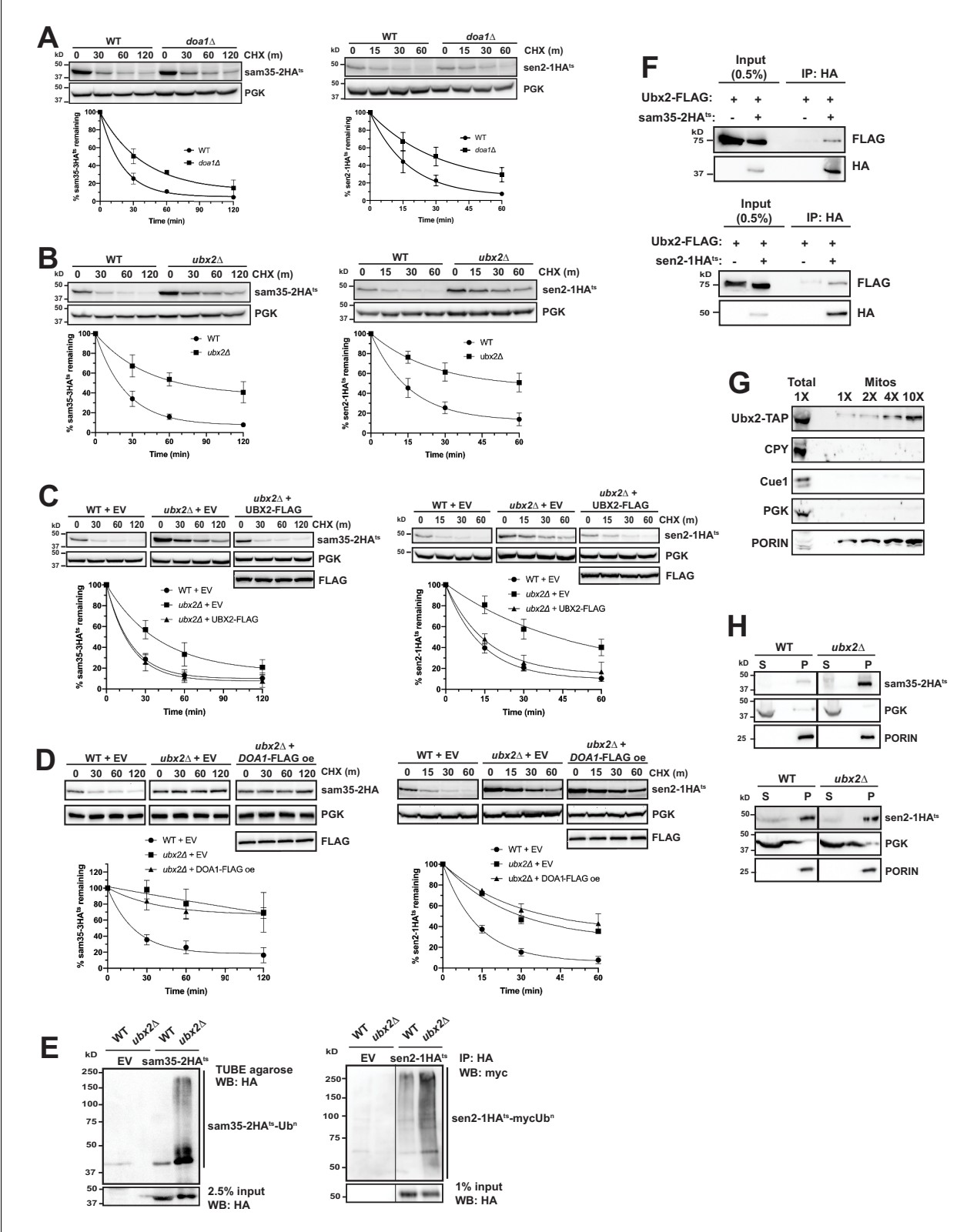

**Figure 6.** The Cdc48 co-factors Ubx2 and Doa1 are implicated in MAD. (**A**) CHX chase for the indicated times at 37°C assessing the turnover of sam35-2HA[ts] (pMM157) and sen2-1HA[ts] (pMM160) in WT (BY4741) and *doa1Δ* cells (yJS208). Proteins were detected by immunoblotting. Graphed below is the mean and SD of the PGK-normalized HA signal at each time point for three biological replicates. (**B**) CHX chase as in A for the indicated times at 37°C assessing the turnover of sam35-2HA[ts] (pMM157) and sen2-1HA[ts] (pMM160) in WT (BY4741) and *ubx2Δ* cells (yJS155). (**C**) CHX chase as in A for the

*Figure 6 continued on next page*

*Figure 6 continued*

indicated times at 37°C assessing the turnover of sam35-2HA$^{ts}$ (pMM157) and sen2-1HA$^{ts}$ (pMM160) in WT (BY4741) and *ubx2Δ* (yJS155) cells co-expressing either empty vector (EV; pRS315) or *CEN* Ubx2-FLAG (pMM242). (**D**) CHX chase as in A for the indicated times at 37°C assessing the turnover of sam35-2HA$^{ts}$ (pMM231) or sen2-1HA$^{ts}$ (pMM234) in WT (BY4741) cells or *ubx2Δ* (yJS155) cells expressing either EV (pRS315) or Doa1-FLAG (pMM254) from a high copy 2μ plasmid. (**E**) Ubiquitination of sam35-2HA$^{ts}$ and sen2-1HA$^{ts}$ was assessed by IP using TUBE agarose or anti-HA agarose from *ubx2Δ* (yJS155) and WT (BY4741) lysates expressing EV (pRS315), sam35-2HA$^{ts}$ (pMM157), or sen2-1HA$^{ts}$ (pMM160), followed by immunoblotting with HA or c-myc antibody. 2.5% or 1% of the IP input lysate was reserved and analyzed by immunoblotting. (**F**) Co-IP of Ubx2-FLAG (pMM242) with sam35-2HA$^{ts}$ or sen2-1HA$^{ts}$ (pMM231 and 234, respectively) from *pre1-1 pre2-2* (WCG4-11/21a) cells was assessed by immunoblotting with the indicated antibodies. IP of Ubx2-FLAG from cells co-expressing EV (pRS316) in place of HA-tagged substrates and 0.5% of the input lysate are shown for comparison. (**G**) Lysate ('Total') and increasing amounts of mitochondria purified by 12,000x*g* and sucrose gradient fractionation ('Mitos') from Ubx2-TAP-expressing cells were examined by immunoblotting with the indicated antibodies. (**H**) Lysates from WT (BY4741) and *ubx2Δ* (yJS155) cells expressing sam35-2HA$^{ts}$ or sen2-1HA$^{ts}$ (pMM157 and 160, respectively) were fractionated at 12,000x*g* at 37°C into mitochondrial pellets (P) and post-mitochondrial supernatants (S). Fractions were subject to immunoblotting with antibodies to HA, PGK, and PORIN.

The online version of this article includes the following source data and figure supplement(s) for figure 6:

**Source data 1.** Quantifications of cycloheximide chases.
**Figure supplement 1.** The Cdc48 co-factors Ubx2 and Doa1 are required for MAD.

2HA$^{ts}$ and sen2-1HA$^{ts}$ degradation, we assessed their turnover in mutants of the well-characterized ER-mitochondrial encounter structure (ERMES) complex. Loss of individual ERMES subunits reduces ER-mitochondrial tethering by greater than 70% and partially disrupts ion and lipid exchange between the organelles (*Kornmann and Walter, 2010*; *Lahiri et al., 2014*; *Murley and Nunnari, 2016*). The turnover of sam35-2HA$^{ts}$ and sen2-1HA$^{ts}$ was unaffected in strains mutant for each of the four ERMES components (*Figure 6—figure supplement 1J*), indicating that significant reductions in ER-mitochondrial contact do not impair their degradation. Furthermore, ts- protein degradation is unaffected by combined loss of the major ERAD E2s, Ubc6 and Ubc7, or E3s, Doa10 and Hrd1 (*Figure 6—figure supplement 1K and L*), with which Ubx2 functionally and physically interacts at the ER (*Neuber et al., 2005*; *Schuberth and Buchberger, 2005*). Ubc6, Ubc7, and Doa10 have also been implicated in the degradation of mislocalized tail-anchored proteins subsequent to Msp1-dependent removal from mitochondria (*Dederer et al., 2019*; *Matsumoto et al., 2019*). Degradation of sam35-2HA$^{ts}$ and sen2-1HA$^{ts}$is independent of Msp1 (*Figure 5—figure supplement 1A*). Finally, as only ~20% of the ER is found in the mitochondrial pellet fraction at 12,000x*g* (*Figure 2—figure supplement 1D*), the ts- substrates would be expected to accumulate predominantly in the post-mitochondrial supernatant (S) if they were being degraded from the ER. However, fractionation data in the proteasome mutant strain (*Figure 2C* and *Figure 2—figure supplement 1E*) strongly suggests that ts- protein degradation occurs from the mitochondria. Sam35-2HA$^{ts}$ and sen2-1HA$^{ts}$ also largely accumulate in the mitochondrial pellet in *ubx2Δ* cells (*Figure 6H*). Similar to the ts- proteins, degradation of Fzo1HA, which is a substrate for the cytosolic E3 SCF$^{MDM30}$ and its cognate E2 Cdc34 (*Cohen et al., 2008*), is unaffected by loss of ERMES components (*Figure 6—figure supplement 1M*). Fzo1 also accumulates predominantly in the mitochondrial pellet when Ubx2 is absent (*Figure 6—figure supplement 1N*). Thus, our data indicate that the mitochondrial population of Ubx2 interacts with UPS-targeted MAD substrates post-ubiquitination to facilitate their degradation from the MOM.

## Discussion

Mitochondria are essential to cellular bioenergetics and metabolism. It is therefore vital that QC mechanisms dispose of damaged proteins that can compromise function. In mammalian cells, this is accomplished at the macroscopic level through mitophagy (*Pickles et al., 2018*). Here, we define a UPS-mediated pathway for the degradation of individual dysfunctional yeast MOM proteins using two newly-established model QC substrates, sam35-2HA$^{ts}$ and sen2-1HA$^{ts}$ (schematized in *Figure 7*). While previous mitochondrial UPS substrates have consisted primarily of native proteins (*Belgareh-Touzé et al., 2017*; *Cohen et al., 2008*; *Heo et al., 2010*; *Wu et al., 2016*), sam35-2HA$^{ts}$ and sen2-1HA$^{ts}$ contain mutations that render them unstable at the non-permissive temperature, and thus mimic damaged, misfolded proteins. These new mitochondrial model substrates have revealed a

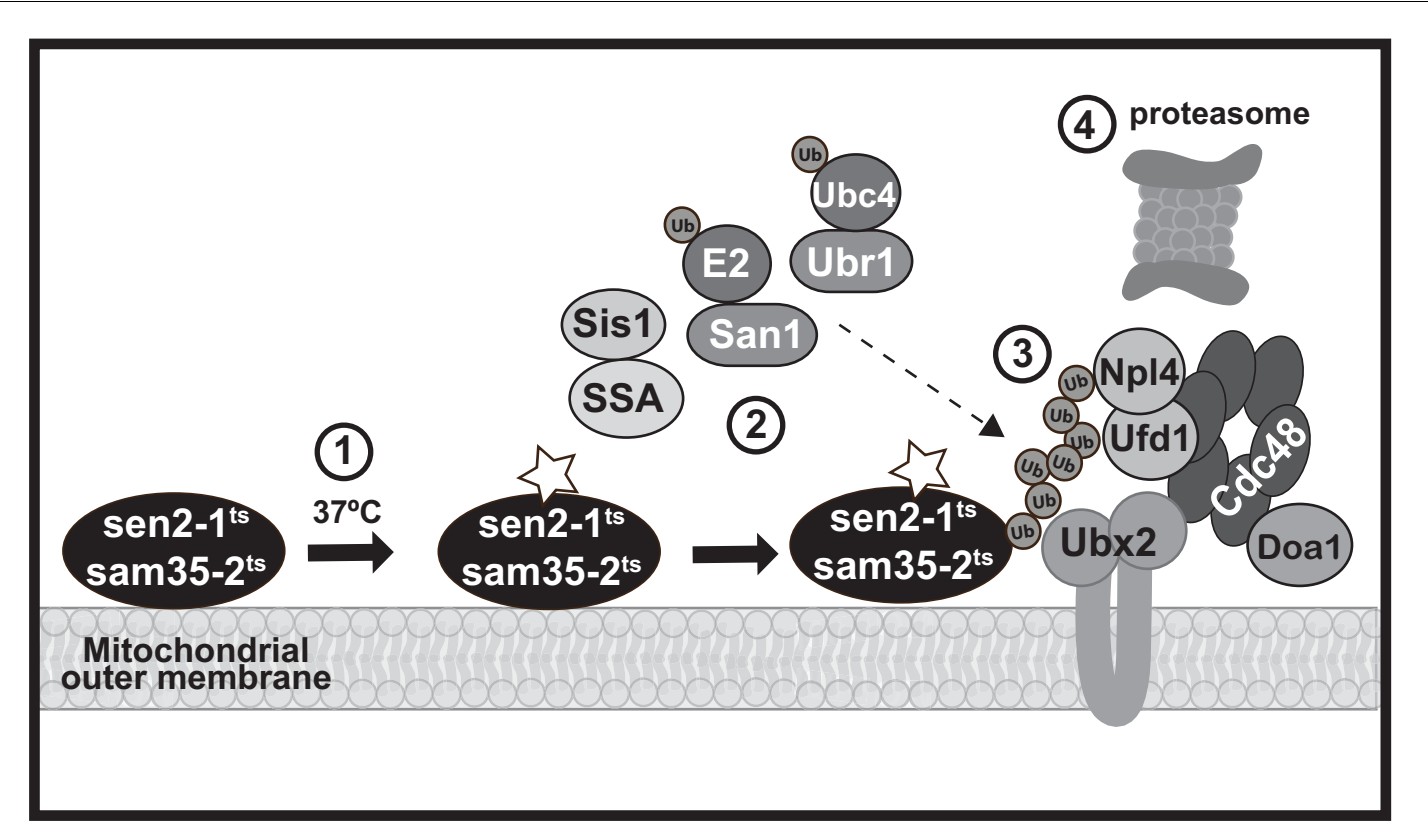

**Figure 7.** A model MAD QC pathway based on the present study. When the temperature is increased to 37°C, the peripheral MOM ts- proteins sam35-2HA[ts] and sen2-1HA[ts] become non-functional (denoted by a star) yet remain at the mitochondrial outer membrane (step 1). They are recognized as quality control substrates and ubiquitinated (step 2), which requires cytosolic chaperones (Ssa1 and Sis1) and the ubiquitin ligase San1 (for sam35-2HA[ts]) or Ubr1 and the ubiquitin conjugating enzyme Ubc4 (for sen2-1HA[ts]). Once ubiquitinated, the Cdc48-Npl4-Ufd1 unfoldase, along with its co-factor Doa1 and a mitochondria-localized pool of its co-factor Ubx2 (step 3), act to direct them to the 26S proteasome for degradation (step 4).

tightly-coupled degradation pathway at the MOM that requires both cytosolic and mitochondrial machinery.

This MAD QC pathway utilizes factors that have not been previously linked to mitochondria or implicated in MAD of native MOM proteins. It is also not identical to any other cellular QC pathway, although there are points of intersection. All of these degradation pathways, unsurprisingly, require 26S proteasome activity. In many cases, the involvement of the Cdc48-Npl4-Ufd1 AAA-ATPase complex represents another point of convergence (*Benischke et al., 2014*; *Cohen et al., 2008*; *Gallagher et al., 2014*; *Heo et al., 2010*; *Jarosch et al., 2002*; *Tanaka et al., 2010*; *Wu et al., 2016*; *Xu et al., 2011*). However, tail-anchored proteins mistargeted to mitochondria and unimported mitochondrial precursors that accumulate at the mitochondrial surface following mitochondrial import stress also require the AAA-ATPase Msp1 for their recognition and extraction from mitochondria (*Chen et al., 2014*; *Matsumoto et al., 2019*; *Okreglak and Walter, 2014*; *Weidberg and Amon, 2018*; *Wohlever et al., 2017*). We found Msp1 to be dispensable for the degradation of mitochondrial ts- substrates. It was recently determined that the degradation of these mislocalized tail-anchored proteins involves their re-localization to the ER following Msp1 extraction, where they are then ubiquitinated by ERAD machinery (*Dederer et al., 2019*; *Matsumoto et al., 2019*). The MAD QC pathway characterized here is distinct from this pathway: ts-substrates remain mitochondrial and are degraded independently of ERAD E2s or E3s. The molecular determinants, spatial restrictions, and co-factors that underlie ubiquitination and degradation at the mitochondria versus Msp1-dependent ER re-targeting will be of particular interest to elucidate going forward.

The Cdc48-Npl4-Ufd1 complex plays a role in both of these pathways and utilizes unique substrate recruitment co-factors. Of these co-factors, Vms1 is recruited to mitochondria from the cytosol in response to translational or oxidative stress (*Izawa et al., 2017*; *Nielson et al., 2017*) and cytosolic Doa1 is implicated in the turn-over of native MOM proteins (*Wu et al., 2016*). Only loss of Doa1 had an effect on mitochondrial ts- protein degradation. However, we found another Cdc48 complex co-factor, Ubx2, to have a substantially greater role in degradation of the two ts- substrates. Ubx2 is well-known as an ER transmembrane protein with a role in ERAD. Here, we provide strong evidence for a discrete, functional, mitochondrial pool of Ubx2. Interestingly, we find that Ubx2 also interacts with Fzo1 at mitochondria and is required for its degradation. This raises the interesting possibility that Ubx2, as well as its mammalian ortholog UbxD8, will have a broad role in MAD and mitochondrial homeostasis in addition to its role in ERAD. Consistent with this, while this manuscript was in preparation, a role for Ubx2 in the turnover of mitochondrial precursor proteins arrested in the Tom40 translocon was reported (*Mårtensson et al., 2019*).

With regard to ubiquitination, we find no evidence for the involvement of Rsp5 or SCF$^{Mdm30}$ ubiquitin ligases, which are both implicated in ubiquitination of native MOM proteins and/or maintenance of mitochondrial integrity (*Belgareh-Touzé et al., 2017*; *Cohen et al., 2008*; *Escobar-Henriques et al., 2006*; *Fritz et al., 2003*; *Wu et al., 2016*). Instead, San1 and Ubr1, which have broad roles as quality control E3s for misfolded cytosolic and nuclear proteins (*Amm et al., 2016*; *Amm and Wolf, 2016*; *Eisele and Wolf, 2008*; *Gardner et al., 2005*; *Guerriero et al., 2013*; *Heck et al., 2010*; *Khosrow-Khavar et al., 2012*; *Lewis and Pelham, 2009*; *Nillegoda et al., 2010*; *Prasad et al., 2012*; *Prasad et al., 2018*; *Samant et al., 2018*; *Summers et al., 2013*), are required to degrade mitochondrial ts- substrates. Recent studies suggest that the degradation of many cytosolic San1 and Ubr1 substrates requires prior nuclear import (*Prasad et al., 2018*; *Samant et al., 2018*). Sam35-2HA$^{ts}$ and sen2-1HA$^{ts}$, however, remain mitochondrial when degradation is blocked and specific factors implicated in nuclear import were found to be dispensable. While there have also been reports of Ubr1 contributing to ERAD (*Stolz et al., 2013*), as noted above, MAD QC appears to be distinct from ERAD.

For ubiquitination of QC substrates to occur, they must first be recognized as being improperly folded. In most QC systems, this recognition requires chaperones. Here, the involvement of the SSA family of Hsp70s in mitochondrial ts- protein degradation reflects commonality with cytosolic QC pathways and ERAD pathways for cytosolic misfolded domains (ERAD-C). The SSA family of chaperones is required for co-translational folding or import of at least some mitochondrial proteins (*Ben-Menachem et al., 2018*; *Deshaies et al., 1988*; *Sass et al., 2003*; *Young et al., 2003*), positioning them to play a role in recognizing misfolded mitochondrial proteins and targeting them to the UPS. On the other hand, we have found that the degradation of a native MOM protein, Fzo1, largely does not require Hsp70 chaperones, extending to mitochondria a distinction between misfolded substrates and those whose ubiquitination occurs in a regulated manner.

Whether there are other, yet to be identified, factors involved in the degradation of MOM proteins remains to be seen. Several mitochondrial inner membrane and IMS proteins have also been identified as proteasome substrates (*Bragoszewski et al., 2013*; *Margineantu et al., 2007*; *Pearce and Sherman, 1997*; *Radke et al., 2008*). It now becomes of interest to ascertain how their degradation overlaps with and diverges from the QC pathway defined herein. Finally, it will also be important to determine how our findings extend to mammalian mitochondrial protein turnover and mitochondrial homeostasis as a whole.

## Materials and methods

### Yeast strains, plasmids, and growth conditions

*Saccharomyces cerevisiae* strains expressing ts- alleles were cultured at 25°C in minimal media supplemented with 2% glucose and the appropriate amino acids, unless otherwise indicated. For spot growth assays, 10-fold serial dilutions beginning with 0.1 OD$_{600}$ units of cells were spotted to YPD and incubated at 25°C for 3 days or 37°C for 2 days.

Strains used in this study can be found in the *Supplementary file 1*: Key Resources Table. Deletion collection strains were confirmed by PCR using a KanMX-specific oligo (oMM19; see *Supplementary file 1*: Key Resources Table for oligo sequences) paired with ORF-specific primers

annealing 500 bp upstream of the start codon. A strain expressing genomic sam35-2HA[ts] (yMM37) was constructed by a one-step PCR-mediated HA tagging using pFA6a-3HA-His3M×6 (*Longtine et al., 1998*) as a PCR template with oligos oMM84 and oMM85 and integration into a sam35-2[ts] strain (*Li et al., 2011*). Deletion strains yMM149 (ubr1::KanMX) and yJS155 (ubx2::KanMX) were constructed by PCR-mediated gene disruption using yeast deletion collection strains (GE Dharmacon) as PCR templates with oligos oMM236 and oMM237, followed by integration into strain WCG4a (for yMM149) or oligos oJS18 and oJS20 with integration into BY4741 (for yJS155). Strain yJS208 (doa1::KanMX) was constructed by PCR-mediated gene disruption using pFA6a-KanMX6 (*Longtine et al., 1998*) as a PCR template with oligos oMM257 and oMM258 and integration in strain WCG4a.

Plasmids used in this study can be found in the *Supplementary file 1*: Key Resources Table. Plasmid pMM157 was constructed by PCR amplification of sam35-2HA[ts] with adjacent promoter and terminator sequence from yMM37 using oligos oMM128 and oMM129 containing flanking XhoI and XbaI restriction sites, respectively, and ligation of this insert into the XhoI and XbaI sites in pRS315 (*Sikorski and Hieter, 1989*). Plasmid pMM160 was constructed by PCR amplification of sen2-1HA[ts] with adjacent promoter and terminator sequence from yMM41 using oligos oMM129 and oMM130 containing flanking XbaI and XhoI restriction sites, respectively, and ligation of this insert into the XhoI and XbaI sites in pRS315. Plasmids pMM231 and 234 were constructed by subcloning the Xho1/XbaI flanked insert from pMM157 and 160, respectively, into pRS316 (*Sikorski and Hieter, 1989*). Plasmids pMD1 and pMD4 were constructed by digestion of pMM157 and pMM160, respectively, with BamHI/AscI to replace the HA tag with GFP from pFA6a-GFP cut with the same sites.

Plasmid pMM242 was generated in two steps. First, pUBX2-UBX2-TAP with flanking XhoI and AscI sites was PCR amplified using a Ubx2-TAP strain (GE Dharmacon) as a template with oligos oJS18 and oJS19 and ligated into pRS315. The TAP tag was then dropped out by restriction digest with BamHI/AscI and replaced with annealed oligos oMM240 and oMM241 encoding a 3x FLAG epitope with overhang compatible for ligation into BamHI/AscI. pMM254 was generated by PCR amplifying the Doa1 promoter and ORF from the genome of BY4741 with oligos containing flanking XhoI (oMM267) and AscI/3xFLAG epitope (oMM268) and ligation into pRS426 (*Christianson et al., 1992*) digested at the same sites.

## Antibodies

Rabbit polyclonal anti-Sam35 (*Chan and Lithgow, 2008*) was a generous gift from Trevor Lithgow. Rabbit polyclonal anti-Sis1 (*Yan and Craig, 1999*) was a generous gift from Elizabeth Craig. Rabbit polyclonal anti-Cue1 and anti-ubiquitin were described previously (*Kostova et al., 2009*). Commercial antibodies used were: mouse monoclonal PORIN (MTCO1; abcam); rabbit polyclonal Prc1 (CPY; abcam); mouse monoclonal GFP (Santa Cruz Biotechnology); mouse monoclonal phosphoglycerate kinase 1 (PGK; Life Technologies); rat monoclonal peroxidase-conjugated anti-HA (3F10; Roche); mouse monoclonal anti-FLAG (M2; Sigma-Aldrich); rabbit polyclonal anti-FLAG (Sigma-Aldrich); and rabbit polyclonal anti-c-myc (Abcam).

## Cycloheximide chase and immunoblotting

Cycloheximide (CHX) chase analyses were performed as described previously *Metzger et al. (2013)* at 25°C or 37°C. For chases at 37°C, cells were cultured at the permissive temperature of 25°C until the addition of 100 μg/mL CHX, after which the temperature was increased to 37°C to accelerate the turnover of the ts- proteins. Maintaining cells at 25°C until the addition of CHX may be insufficient to fully inactivate ts- conditional yeast strains prior to the CHX chases, resulting in an underestimation of the role of the inactivated protein in degradation of substrates. Cells were then lysed in 1% β-mercaptoethanol (βme)/250 mM NaOH and proteins were precipitated in 5% trichloroacetic acid (TCA). Protein pellets were resuspended in TCA sample buffer (3.5% SDS, 0.5 M DTT, 80 mM Tris pH8.8, 8 mM EDTA, 15% glycerol, 0.1 mg/mL bromophenol blue). Samples were analyzed by SDS-PAGE and immunoblotting with the indicated antibodies according to standard procedures. Proteins were detected using SuperSignal West Pico Luminol Enhancer Solution (Thermo Scientific) or Amersham ECL Select (GE Healthcare) and a G:box (Syngene) or c280 Imager (Azure). Each CHX chase was repeated at least three times; shown in each figure is a representative blot. The percent of substrate remaining was calculated by quantification of anti-HA signal using ImageJ (National

Institutes of Health, Bethesda, MD), followed by normalization of this signal to the anti-PGK signal at the same time point. The '0' time points were set to 100% and the mean and standard deviation (SD) at each time point for three independent biological replicates were graphed using GraphPad Prism 8 and fitted with one phase decay curves. Time points appearing to not have error bars have SD smaller than the size of the symbol.

## Mitochondrial fractionation and sodium carbonate extraction

Mitochondria were isolated as described previously *Gregg et al. (2009)*, with the following changes for the analysis of ts- proteins. Cultures were grown in minimal media containing the appropriate amino acids at 25°C. Cells were incubated in DTT Buffer (*Gregg et al., 2009*) for 30 min at 25°C and in Zymolyase Buffer (*Gregg et al., 2009*) with Zymolyase-100T (MP Biomedicals) for 45 min at 25°C, after which time the resulting spheroplasts were washed and resuspended in an equal volume of Zymolyase Buffer without Zymolyase-100T, either at 25°C or prewarmed to 37°C, as indicated. CHX (100 µg/mL) was added and spheroplasts were incubated without shaking at 25°C or 37°C for 5 min (sen2-1HA$^{ts}$) or 15 min (sam35-2HA$^{ts}$). Spheroplasts were then homogenized using a glass homogenizer and the resulting cleared lysate was fractionated at 12,000x$g$ into a post-mitochondrial supernatant fraction (S) and mitochondrial pellet (P). Both fractions were precipitated in 10% TCA and washed in 2% TCA prior to resuspension in TCA sample buffer. Equivalent proportions of S and P were analyzed by SDS-PAGE and immunoblotting.

For the isolation of purified mitochondria devoid of other organelles, mitochondrial pellets isolated as above were resuspended in 3 mL SEM buffer (10 mM Tris-HCl pH 7.4, 0.6 M sorbitol, 1 mM EDTA, 0.2% BSA) and overlaid on a sucrose gradient layered top to bottom with 6 mL 15% (weight/volume) sucrose/6 mL 23% sucrose/16 mL 32% sucrose/6 mL 60% sucrose and spun in a swinging-bucket rotor at 134,000x$g$ for 1 hr at 4°C. The intact mitochondria residing at the 60%/32% sucrose interface were recovered and gently resuspended in SEM buffer and spun in a swinging-bucket rotor at 10,000x$g$ for 30 min at 4°C. The pure mitochondrial pellet was precipitated in 10% TCA and washed in 2% TCA prior to resuspension in TCA sample buffer.

For sodium carbonate ($Na_2CO_3$) extraction of peripheral mitochondrial proteins, crude mitochondria isolated as above were treated with 0.2 M $Na_2CO_3$ or NaCl as described previously *Boldogh and Pon (2007)* and analyzed by immunoblotting.

## Ubiquitin immunoblotting and tandem ubiquitin-binding entities (TUBE) isolation of ubiquitinated proteins

For ubiquitin visualization using anti-ubiquitin immunoblotting, 10–30 OD$_{600}$ units of mid log phase (OD$_{600}$ = 0.8–1) cells were grown at 25°C and incubated at 37°C for 30 min prior to protein preparation using βme/NaOH/TCA, as described above. Protein pellets resuspended in TCA sample buffer were diluted in Dilution buffer (50 mM Tris-HCl pH 7.5, 100 mM NaCl, 5 mM EDTA, 5% glycerol, 1% Triton X-100, 1x Complete Protease Inhibitor Cocktail (Roche), 1 mM NEM) and HA-tagged substrates were isolated by immunoprecipitation (IP) for 18 hr at 4°C using mouse monoclonal anti-HA affinity matrix (Sigma). Immunoprecipitated proteins were eluted with SDS-PAGE sample buffer and analyzed by SDS-PAGE. Ubiquitinated species were visualized by immunoblotting with ubiquitin antibodies. Unmodified substrates were detected by immunoblotting with HA antibodies.

For ubiquitin visualization using anti-myc immunoblotting, 10–30 OD$_{600}$ units of early log phase (OD$_{600}$ = ~0.4) cells were grown at 25°C, treated with 0.1 mM CuSO$_4$ for 4 hr to induce expression of myc-ubiquitin, then incubated at 37°C for 30 min. Proteins were prepared as above. Mouse monoclonal anti-HA affinity matrix was pre-blocked with 1% ovalbumin and protein lysate was pre-cleared with Glutathione Sepharose 4-B (GE Healthcare Life Sciences) overnight at for 18 hr at 4°C, prior to IP of cleared protein lysate with the pre-blocked anti-HA affinity matrix for 4 hr at 4°C. Ubiquitinated species were visualized by immunoblotting with rabbit anti-c-myc antibodies. Unmodified substrates were detected by immunoblotting with HA antibodies.

For TUBE isolation of ubiquitinated proteins, 30–100 OD$_{600}$ units of mid log phase cell were grown at 25°C and incubated at 37°C for 30 min, at which time they were frozen at −80°C. Cell lysis was performed using glass beads in Lysis buffer (50 mM Tris-HCl pH 7.5, 200 mM NaCl, 1% Triton X-100, 1x Complete Protease Inhibitor Cocktail, 1 mM NEM). Lysates were cleared by centrifugation at 13,000x$g$ for 20 min, then centrifugation at 13,000x$g$ for 5 min. An aliquot of input lysate was

precipitated in 10% TCA and protein pellets were resuspended in TCA sample buffer. The remainder of the lysate was incubated with TUBE-1 agarose (Life Sensors) for 18 hr at 4˚C; ubiquitinated proteins were eluted from the TUBE-1 agarose with SDS-PAGE sample buffer. HA-tagged substrates in the input and TUBE-bound species were analyzed by SDS-PAGE and immunoblotting with HA antibodies.

For analysis of ubiquitination from 12,000xg mitochondria and post-mitochondrial supernatant fractions, cells were grown at 25˚C and treated with 0.1 mM $CuSO_4$ for 2 hr to induce expression of myc-ubiquitin prior to crude mitochondrial fractionation as described in the previous section. S and P fractions were then subject to protein preparation and IP as described above for ubiquitin visualization using anti-myc immunoblotting.

## Co-immunoprecipitation

For co-IP, 10–30 $OD_{600}$ units of mid log phase ($OD_{600}$ = 0.8–1) cells expressing Ubx2-FLAG were grown at 25˚C and incubated at 37˚C for 1 hr prior to glass bead lysis in Lysis Buffer (50 mM Tris-HCl pH7.5, 100 mM NaCl, 5% glycerol, 1 mM DTT, 1 mM PMSF, 1x Complete Protease Inhibitor Cocktail, 1 mM NEM). Cleared lysate was added to anti-HA affinity matrix and immunoprecipitated for 18 hr at 4˚C; bound proteins were eluted with SDS-PAGE sample buffer and analyzed by immunoblotting.

## Microscopy

For imaging Ubx2-GFP, logarithmically-growing yeast cells were immobilized on coverslips coated with concanavalin A and imaged using a Nikon Eclipse Ti inverted microscope, equipped with a 64 μm pixel CoolSNAP $HQ^2$ camera (Photometrics), Intensilight C-HGFIE illuminator, and 100x NA 1.42 Plan Apo objective. For live-cell analysis of ts- proteins, logarithmically growing cells were embedded in agarose and treated with 100 μg/mL CHX at 25˚C (time = 0) and then incubated at 37˚C for the indicated times before imaging using a Nikon Eclipse Ti inverted microscope u60x NA 1.45 Plan Apo objective, Yokogawa spinning disc, 488 and 561 nm excitation lasers (Agilent technology MCL-400), back-illuminated EMCCD camera (Andor, DU888), and a 2x relay lens. A Nikon DS-U3 camera was used to record DIC images. 200 nm thick Z-sections spanning entire cell (~6 μm), were acquired. ImageJ (National Institutes of Health) was used to assemble the figures.

## Acknowledgements

We would like to thank Charlie Boone, Mickael Cohen, Elizabeth Craig, Mark Hochstrasser, Trevor Lithgow, Susan Michaelis, Benedikt Westermann, and Dieter Wolf for their generous gifts of strains, antibodies, and plasmids; We thank Jeffrey Brodsky, Mickael Cohen, Susan Michaelis, and Yien Che Tsai for helpful discussion and comments regarding this manuscript. This research was supported by the Intramural Research Program of the NIH, National Cancer Institute, Center for Cancer Research.

## Additional information

### Funding

| Funder | Grant reference number | Author |
|---|---|---|
| National Cancer Institute | Intramural Research Program | Allan M Weissman |

The funders had no role in study design, data collection and interpretation, or the decision to submit the work for publication.

### Author contributions

Meredith B Metzger, Conceptualization, Data curation, Formal analysis, Supervision, Investigation, Visualization, Methodology, Project administration; Jessica L Scales, Validation, Investigation, Visualization, Methodology; Mitchell F Dunklebarger, Investigation; Jadranka Loncarek, Resources, Investigation, Visualization, Methodology; Allan M Weissman, Conceptualization, Supervision, Methodology, Project administration

## Author ORCIDs

Meredith B Metzger [iD] https://orcid.org/0000-0002-6248-0009
Allan M Weissman [iD] https://orcid.org/0000-0002-7865-7702

## Decision letter and Author response

Decision letter https://doi.org/10.7554/eLife.51065.sa1
Author response https://doi.org/10.7554/eLife.51065.sa2

## Additional files

### Supplementary files

- Supplementary file 1. Key resources table.

- Transparent reporting form

### Data availability

All data generated of analyzed during this study are included in the manuscript and supporting files. Source data for all figure quantifications have been provided.

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
