## [Decision Letter]

**Acceptance summary:**

The mitochondria-associated degradation (MAD) pathway mediates ubiquitination and removal of mitochondrial outer membrane (MOM) proteins, and targets them for degradation by the proteasome. So far, the components and the mechanism of this process have been poorly described. In this paper, the authors report the identification of components of this degradation pathway and uncover mechanisms in actions, providing an important contribution to the understanding of protein quality control pathways operating in the cell.

**Decision letter after peer review:**

Thank you for submitting your article "A protein quality control pathway at the mitochondrial outer membrane" for consideration by *eLife*. Your article has been reviewed by three peer reviewers, and the evaluation has been overseen by a Reviewing Editor and David Ron as the Senior Editor. The following individual involved in review of your submission has agreed to reveal their identity: Sonya Neal (Reviewer #3).

The reviewers have discussed the reviews with one another and the Reviewing Editor has drafted this decision to help you prepare a revised submission.

The study reveals a new pathway of quality control for mitochondrial outer membrane proteins that involves E ligases, other key and adaptor proteins as well as cytosolic chaperones. The pathway serves to remove temperature-destabilized variants of mitochondrial proteins.

The reviewers agree that the study is interesting and in principle well done. The findings are an important contribution to the field. However, several additions and controls are needed to bring this interesting story to *eLife*'s standards.

The data concerning the involvement of E3 ligases, Cdc48, Sis1 and Doa1 should be strengthened. The efforts in this direction should involve an appropriate statistics of repetitions, and should include a clear explanation concerning high molecular weight bands (hardly entering the gel). Furthermore, in order to validate that ubiquitination is dependent on a substrate being misfolded, the ubiquitination status of wild type versions of Sam35 and Sen2 should be assessed as well for comparison to their mutant allele counterparts.

Along the similar lines of testing wild-type proteins and their accessibility for this quality control pathway, it would be good to know whether Sam35 and Sen2 bind to Ubx2 as well or whether this interaction is specific for misfolded proteins.

Second, it is important to consider and address a possibility of substrates transfer to the ER membrane, published recently by the Endo lab. The analyses should include a systematic estimation of ER contaminations in the fractionations, or look at a putative relocation by some other means, i.e. microscopy, especially under conditions of degradation block/inhibition.

Third, to provide important mechanistic facts, the authors should determine the localization of the ubiquitinated molecules (and not only the non-modified species – Figure 3F and G). Are the Ub molecules enriched in the soluble fraction or still remain associated with mitochondria?

[Editors' note: further revisions were suggested prior to acceptance, as described below.]

Thank you for resubmitting your work entitled "A protein quality control pathway at the mitochondrial outer membrane" for further consideration by *eLife*. Your revised article has been evaluated by David Ron as the Senior Editor, a Reviewing Editor and three peer reviewers.

The manuscript has been improved but there is one remaining issue that needs to be addressed before acceptance. In new Figure 6—figure supplement 1D upper panel, the HA-antibody recognizes a prominent band in the absence of HA-tagged Sam35 (upper panel, lane 1); please comment on the nature of this band.

---

## [Author Response]

The reviewers agree that the study is interesting and in principle well done. The findings are an important contribution to the field. However, several additions and controls are needed to bring this interesting story to eLife's standards.The data concerning the involvement of E3 ligases, Cdc48, Sis1 and Doa1 should be strengthened. The efforts in this direction should involve an appropriate statistics of repetitions…

In the revised manuscript, we have included graphs for cycloheximide chase experiments in Figures 1-6 showing the mean exponential decay of the ts- substrates, as quantified from at least 3 biological replicates, with error bars showing standard deviation.

… and should include a clear explanation concerning high molecular weight bands (hardly entering the gel).

We believe the reviewers are referring to the higher molecular weight signal accumulating at the top of some of our blots of ubiquitinated material (i.e. Figure 3C). It is not uncommon for in vivo highly ubiquitinated species to migrate very slowly into Tris-Glycine acrylamide gels. A recent review on the analysis of ubiquitin chains by immunoblotting highlights this issue: “Proteins can be modified by 20 or more ubiquitin molecules that can add >200 kDa to their molecular mass…” (Emmerich and Cohen, 2015), We have included an explanation for this and referenced Emmerich and Cohen in the last paragraph of the subsection “The degradation of MAD QC substrates requires the ubiquitin-proteasome system”.

Furthermore, in order to validate that ubiquitination is dependent on a substrate being misfolded, the ubiquitination status of wild type versions of Sam35 and Sen2 should be assessed as well for comparison to their mutant allele counterparts.

We have assessed the ubiquitination status of SEN2HA and SAM35HA and find their ubiquitination to be significantly less than their ts- counterparts. These data are now included in Figure 2—figure supplement 1G and mentioned in the last paragraph of the subsection “The degradation of MAD QC substrates requires the ubiquitin-proteasome system”.

Along the similar lines of testing wild-type proteins and their accessibility for this quality control pathway, it would be good to know whether Sam35 and Sen2 bind to Ubx2 as well or is this interaction specific for misfolded proteins.

Loss of Ubx2 does not affect the turnover of either SAM35HA or SEN2HA, however these proteins are quite stable to begin with. We have included these data in Figure 6—figure supplement 1C and in the text. While we detect some interaction between the WT versions of the ts- proteins and Ubx2-FLAG, the WT proteins are also much more abundant than the ts- proteins, making it difficult to draw a strong conclusion regarding this interaction. We have included these new data in Figure 6—figure supplement 1D and in the third paragraph of subsection “The Cdc48 complex co-factors Ubx2 and Doa1 are required for MAD”. It is worth noting that we also detect an interaction between Fzo1HA and Ubx2-FLAG (Figure 6—figure supplement 1G), so it seems likely that Ubx2 is able to interact with both misfolded and native ubiquitinated proteins. For clarity, this has been added to the aforementioned paragraph.

Second, it is important to consider and address a possibility of substrates transfer to the ER membrane, published recently by the Endo lab. The analyses should include a systematic estimation of ER contaminations in the fractionations, or look at a putative relocation by some other means, i.e. microscopy, especially under conditions of degradation block/inhibition.

The reviewers bring up a very good point. We assessed ER contamination of our 12,000x*g* S and P fractions and find that the crude mitochondrial pellet (P12,000xg) contains ~20% of the total ER, as assessed by localization of the integral ER membrane protein, Cue1. When we further purify this crude mitochondrial pellet fraction by sucrose gradient fractionation, we can remove around 90% of the contaminating ER, leaving ~2-4% of the total ER remaining in the purified mitochondria. Despite this, we retain ~70% of the peripherally-associated ts- proteins in the purified mitochondria. We think it is important to note (as we now do in the text) that to isolate purified fractions by sucrose gradient, mitochondria are subject to an extensive degree of manipulation. This likely results in the dislodging of some fraction of these peripheral membrane proteins. These experiments were performed in the *pre1-1 pre2-2* proteasome mutant strain, indicating that the majority of the ts- proteins indeed are accumulating at the mitochondria when their degradation is blocked. We have included these new data in Figure 2—figure supplement 1D and E and it is discussed in the second paragraph of the subsection “The degradation of MAD QC substrates requires the ubiquitin-proteasome system”.

Third, to provide important mechanistic facts, the authors should determine the localization of the ubiquitinated molecules (and not only the non-modified species – Figure 3F and G). Are the Ub molecules enriched in the soluble fraction or still remain associated with mitochondria?

We have assessed the ubiquitination of the ts- proteins at the mitochondria and detect ubiquitin modified species there. The distribution observed is similar to that seen in the whole cell lysate, where ubiquitination of the ts- protein is faintly observed in the WT strain and increases in the *pre1-1 pre2-2* strain, all relative to the empty vector control strains. Assessment of ubiquitination in the soluble fractions reveals a lesser portion of ubiquitin-modified protein in the *pre1-1 pre2-2* strain, although it is not clear whether this represents a pool of protein that has dissociated from the mitochondria or is truly cytosolic. We have added these data to Figure 2—figure supplement 1H and discuss them in the last paragraph of the subsection “The degradation of MAD QC substrates requires the ubiquitin-proteasome system”.

[Editors' note: further revisions were suggested prior to acceptance, as described below.]

The manuscript has been improved but there is one remaining issue that needs to be addressed before acceptance. In new Figure 6—figure supplement 1D upper panel, the HA-antibody recognizes a prominent band in the absence of HA-tagged Sam35 (upper panel, lane 1); please comment on the nature of this band.

Please find a corrected version of Figure 6—figure supplement 1 in our revised manuscript. During the revision of the manuscript, we had repeated the experiment in panel D several times. The anti-HA blot shown mistakenly corresponded to an earlier experiment where samples were loaded in a different order. It was an oversight that this portion of the figure was not updated and we have corrected this in the updated figure. We have also checked all of our other new figures to ensure we did not make this mistake elsewhere.

We sincerely apologize for this error and appreciate the careful examination done by the reviewers.